# Stable Port-Hamiltonian Neural Networks

**Fabian J. Roth, Dominik K. Klein, Maximilian Kannapinn, Jan Peters, Oliver Weeger**
Technical University of Darmstadt, Darmstadt, Germany
{roth,klein,kannapinn,weeger}@cps.tu-darmstadt.de,
peters@ias.tu-darmstadt.de

## Abstract

In recent years, nonlinear dynamic system identification using artificial neural networks has garnered attention due to its broad potential applications across science and engineering. However, purely data-driven approaches often struggle with extrapolation and may yield physically implausible forecasts. Furthermore, the learned dynamics can exhibit instabilities, making it difficult to apply such models safely and robustly. This article introduces stable port-Hamiltonian neural networks, a machine learning architecture that incorporates physical biases of energy conservation and dissipation while ensuring global Lyapunov stability of the learned dynamics. Through illustrative and real-world examples, we demonstrate that these strong inductive biases facilitate robust learning of stable dynamics from sparse data, while avoiding instability and surpassing purely data-driven approaches in accuracy and physically meaningful generalization. Furthermore, the model's applicability and potential for data-driven surrogate modeling are showcased on multiphysics simulation data.

## 1 Introduction

Recent years have seen a strongly growing interest in using machine learning (ML) to identify the dynamics of physical systems directly from observations [5, 28, 31, 32, 40, 42, 53]. This approach has diverse applications, ranging from modeling phenomena with partially or entirely unknown underlying physics to creating surrogate and reduced-order models for rapid simulations and digital twins [23, 53, 54]. In the latter application, data-driven methods can address the drawbacks of traditional numerical methods, which often require tremendous computational resources and expertise [1, 53]. However, in turn, classical ML algorithms typically require large amounts of training data and lack robustness, as they often suffer from poor extrapolation and are prone to making physically implausible forecasts [1, 33, 53, 54]. To overcome these limitations, a general consensus is to avoid discarding centuries' worth of established knowledge in physics, which is an implicit consequence of relying solely on data-driven methods. Instead, there is a growing emphasis on incorporating physical priors and inductive biases into ML approaches [33, 54]. This *physics-guided* ML promises improved accuracy, robustness, and interpretability while reducing data requirements [33, 53].

Several physics-guided architectures have been proposed for learning dynamical systems. These approaches often parameterize an ordinary differential equation (ODE) to identify continuous-time dynamics. To ensure adherence to the laws of thermodynamics, many architectures build upon frameworks such as Hamiltonian mechanics [9, 16, 20], Lagrangian mechanics [10, 34], Poisson systems [19], GENERIC [18, 56], port-Hamiltonian systems (PHS) [11, 13, 37, 38, 57], or the generalized Onsager principle [55]. The latter three allow for the modeling of dissipative dynamics, while the former are generally limited to energy-conserving systems. Additionally, PHS explicitly incorporate external excitations, making them particularly suited for modeling controlled systems.

The mentioned physical modeling frameworks have implications for the stability of the learned dynamics. For example, in Hamiltonian neural networks, stable equilibria correspond to extrema in

39th Conference on Neural Information Processing Systems (NeurIPS 2025).

the Hamiltonian energy function. However, these effects are often not discussed in the referenced works. Nevertheless, stability is an important and fundamental property of any dynamic system. Guarantees on the stability of the dynamics may improve the model's robustness, help trust their predictions, and can themselves be viewed as physically motivated biases [14].

Given the importance of the stability property, multiple other works have proposed methods to enhance the stability of learned dynamics, often using neural ordinary differential equations (NODEs) [7] as a basis. Massaroli et al. [36] propose a provably stable NODE variant that guarantees the existence of locally asymptotically stable equilibria. Multiple studies have combined Lyapunov's theory with neural networks (NNs) to guarantee the stability of discrete-time dynamics [14, 29] or to estimate the region of attraction for given dynamical systems [3, 44]. Kolter and Manek [27] describe another Lyapunov-stable architecture based on NODEs. They concurrently learn unconstrained dynamics with a NODE and a convex Lyapunov function. Global stability around an equilibrium point is then ensured by projecting the NODE-dynamics onto the space of stable dynamics as given by the Lyapunov function. Takeishi and Kawahara [50] extend this approach from a stable equilibrium to stable invariant sets. Similar projection-based techniques have also been proposed to learn input-output stable dynamics [26] and dissipative systems [39]. However, these projection methods can introduce discontinuities, hindering training strategies that rely on integrating trajectories rather than matching state-derivative pairs [46]. Other approaches employ physical modeling frameworks to formulate the learning problem directly in the subspace of stable dynamics. Rettberg et al. [43] propose identifying a linear latent PHS, which makes enforcing global stability relatively straightforward. Yu et al. [55] propose a model based on the generalized Onsager principle that can guarantee bounded solutions and the existence of locally asymptotically stable equilibria. While their evolution equation is similar to PHS, their approach does not consider arbitrary, time-dependent external input signals.

This work exploits the connections between stability and physical frameworks and proposes stable port-Hamiltonian neural network (sPHNN) for learning *nonlinear* and *globally stable* dynamical systems. The method is projection-free and can incorporate input signals. We show that the approach is capable of learning equilibrium points, though our numerical evaluations focus on practical applications where the equilibrium is known a priori. In such cases, this prior knowledge can be seamlessly integrated into the proposed architecture, improving prediction accuracy while simultaneously reducing data requirements.

## 2 Background

### 2.1 Stability of dynamical systems

We consider autonomous dynamical systems as first-order ODEs, where the state $\boldsymbol{x}(t) \in \mathbb{R}^n$ evolves according to a vector-valued function $\boldsymbol{f} : \mathbb{R}^n \to \mathbb{R}^n$:

$$\dot{\boldsymbol{x}}(t) = \boldsymbol{f}(\boldsymbol{x}(t)). \tag{1}$$

Suppose $\boldsymbol{f}$ is sufficiently smooth to ensure a unique solution $\boldsymbol{x}(t)$ for any initial condition $\boldsymbol{x}(t_0)$. A solution $\hat{\boldsymbol{x}}(t)$ is *stable* in the sense of Lyapunov if, for any small perturbation in the initial condition, the perturbed solution $\bar{\boldsymbol{x}}(t)$ remains close for all times, that is, if for any $\epsilon > 0$, there exists a $\delta > 0$ such that $\|\hat{\boldsymbol{x}}(t_0) - \bar{\boldsymbol{x}}(t_0)\| < \delta$ implies $\|\hat{\boldsymbol{x}}(t) - \bar{\boldsymbol{x}}(t)\| < \epsilon$ for all $t > t_0$ [35, 52]. A stronger version of stability, namely *asymptotic stability*, is obtained if additionally the perturbed solution $\bar{\boldsymbol{x}}(t)$ converges to $\hat{\boldsymbol{x}}(t)$ for $t \to \infty$ for initial conditions $\|\hat{\boldsymbol{x}}(t_0) - \bar{\boldsymbol{x}}(t_0)\| < \Delta$ with some $\Delta > 0$ [52].

Lyapunov stability is a *local* property, in the sense that for sufficiently large perturbations, the perturbed solution may stray arbitrarily far even from asymptotically stable solutions [49]. Quantifying the permissible extent of perturbations on the initial condition such that the perturbed solution still converges to the unperturbed solution leads to the concept of a *basin of attraction* [30]. Asymptotically stable equilibria, for which the basin of attraction extends to the entire phase space, are called *globally asymptotically stable* [30]. A system with a globally stable equilibrium cannot have any other equilibria, as their existence would imply that there are solutions (namely, the other equilibria) that do not converge to the globally stable equilibrium.

In the following, we consider the case of an equilibrium solution $\boldsymbol{x}(t) = \boldsymbol{0}$, i.e., $\boldsymbol{f}(\boldsymbol{0}) = \boldsymbol{0}$. Placing the equilibrium at the origin is done for ease of notation and is without loss of generality, as arbitrary equilibria can be translated to the origin with a time-independent transformation. The stability of the

equilibrium can be shown through the use of Lyapunov's stability theory by finding a suitable scalar Lyapunov function $V$ [24, 52]. Let $V : D \to \mathbb{R}$ be continuously differentiable and positive definite, that is, $V(\mathbf{0}) = 0$ and $V(\boldsymbol{x}) > 0$ in $D \setminus \{\mathbf{0}\}$, where $D \subseteq \mathbb{R}^n$ is a neighborhood of the origin. If the value of $V$ is nonincreasing along trajectories of the system in $D$, then the equilibrium is stable in the sense of Lyapunov. Mathematically, this is the case if $\dot{V}(\boldsymbol{x})$ is negative semi-definite:

$$\dot{V}(\boldsymbol{x}) = \frac{\partial V}{\partial \boldsymbol{x}}^{\mathsf{T}} \dot{\boldsymbol{x}} = \frac{\partial V}{\partial \boldsymbol{x}}^{\mathsf{T}} \boldsymbol{f}(\boldsymbol{x}) \leq 0 \quad \forall \boldsymbol{x} \in D \setminus \{\mathbf{0}\}. \tag{2}$$

In the case of a negative definiteness ($\dot{V}(\boldsymbol{x}) < 0$ in $D \setminus \{\mathbf{0}\}$), *asymptotic* stability can be concluded. To obtain *global* asymptotic stability, these conditions must be satisfied *globally*, that is, for all $\boldsymbol{x} \in D = \mathbb{R}^n$, *and* the Lyapunov function must be radially unbounded, i.e., $V(\boldsymbol{x}) \to \infty$ for $\|\boldsymbol{x}\| \to \infty$ [24]. The latter condition avoids the existence of solutions that do not converge to the equilibrium by diverging to infinity without violating the decrease condition in Equation (2).

## 2.2 Port-Hamiltonian systems

PHS theory provides a framework for modeling physical systems that generalizes the underlying mathematical structure of Hamiltonian dynamics. It incorporates the modeling of energy-dissipating elements, which are typically absent in classical Hamiltonian systems [51]. Additionally, PHS theory has roots in control theory, emphasizing the interactions of dynamic systems with their environment. In this work, we consider PHS of the form

$$\dot{\boldsymbol{x}} = [\boldsymbol{J}(\boldsymbol{x}) - \boldsymbol{R}(\boldsymbol{x})] \frac{\partial \mathcal{H}}{\partial \boldsymbol{x}}(\boldsymbol{x}) + \boldsymbol{G}(\boldsymbol{x})\boldsymbol{u}(t). \tag{3}$$

Here, $\boldsymbol{x}(t) \in \mathbb{R}^n$ describes the system's state, $\boldsymbol{u}(t) \in \mathbb{R}^m$ its input, and $\mathcal{H} : \mathbb{R}^n \to \mathbb{R}$ denotes the Hamiltonian, which typically represents the total energy of the system [4, 51]. The skew-symmetric *structure matrix* $\boldsymbol{J} : \mathbb{R}^n \to \mathbb{R}^{n \times n}$, $\boldsymbol{J}(\boldsymbol{x}) = -\boldsymbol{J}^{\mathsf{T}}(\boldsymbol{x})$ describes the conservative energy flux within the system. The symmetric positive semi-definite matrix $\boldsymbol{R} : \mathbb{R}^n \to \mathbb{R}^{n \times n}$, $\boldsymbol{R}(\boldsymbol{x}) = \boldsymbol{R}^{\mathsf{T}}(\boldsymbol{x}) \succeq 0$ is called *dissipation matrix* and describes the energy losses. Finally, the *input matrix* $\boldsymbol{G} : \mathbb{R}^n \to \mathbb{R}^{n \times m}$ describes how energy enters and exits the system via the inputs. PHS extend the energy conservation of Hamiltonian systems to the energy dissipation inequality:

$$\dot{\mathcal{H}} = \underbrace{\frac{\partial \mathcal{H}}{\partial \boldsymbol{x}}^{\mathsf{T}} \boldsymbol{J} \frac{\partial \mathcal{H}}{\partial \boldsymbol{x}}}_{=0} - \underbrace{\frac{\partial \mathcal{H}}{\partial \boldsymbol{x}}^{\mathsf{T}} \boldsymbol{R} \frac{\partial \mathcal{H}}{\partial \boldsymbol{x}}}_{\geq 0} + \underbrace{\frac{\partial \mathcal{H}}{\partial \boldsymbol{x}}^{\mathsf{T}} \boldsymbol{G} \boldsymbol{u}(t)}_{s(\boldsymbol{x}, \boldsymbol{u})} \leq s(\boldsymbol{x}, \boldsymbol{u}). \tag{4}$$

In the language of system theory, the system is dissipative with respect to the supply rate $s(\boldsymbol{x}, \boldsymbol{u})$. The formal definition of dissipativity requires $\mathcal{H} \geq 0$, which will be ensured later. Since the supply rate $s(\boldsymbol{x}, \boldsymbol{u})$ is linear in the input $\boldsymbol{u}$, it vanishes for zero-input $s(\boldsymbol{x}, \mathbf{0}) = 0$. The inequality thus ensures that the system cannot gain energy in the absence of an external excitation ($\boldsymbol{u} = \mathbf{0}$). Besides this physical motivation, the dissipativity of PHS plays a crucial role in stability analysis. For the unforced system with $s = 0$, the Hamiltonian inherently fulfills the decrease condition $\dot{\mathcal{H}} \leq 0$. Thus, by using $\mathcal{H}$ as a Lyapunov function, it can be shown that any strict minimum in the Hamiltonian implies a stable equilibrium of the unforced dynamics [51]. Furthermore, the equilibrium is asymptotically stable if the system continuously dissipates energy ($\boldsymbol{R} \succ 0$) for states near the minimum.

The use of the port-Hamiltonian formulation as a basis for an ML model has multiple favorable consequences. When learning system dynamics from data, the structure of Equation (3) provides both a physical bias and the opportunity for interpreting the resulting trained model, while enabling the modeling of a wide range of relevant systems. Equation (4) highlights the distinctive roles played by the three matrix-valued functions. The structure matrix $\boldsymbol{J}(\boldsymbol{x})$ results in conservative dynamics, the dissipation matrix $\boldsymbol{R}(\boldsymbol{x})$ contributes only to the dissipative dynamics, and the input matrix $\boldsymbol{G}(\boldsymbol{x})$ captures the energy flow over the system boundaries due to the inputs. This knowledge improves the interpretability of the trained model components. Furthermore, it allows new ways of incorporating prior knowledge into the system. For example, the learned system can be constrained to be conservative by fixing $\boldsymbol{R} = \mathbf{0}$. Furthermore, the connection of PHS to the decrease condition from Lyapunov's theory provides the opportunity to introduce further desirable stability constraints.

To illustrate the above concepts with a simple example, consider the dynamics of a three-dimensional spinning rigid body described by Euler's rotation equations. Expressing the angular velocity vector

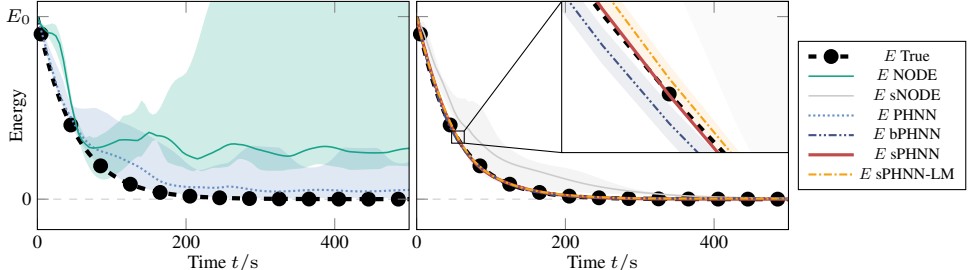

Figure 1. *Spinning rigid body*: Interquartile mean (lines) and range (shaded regions) of the energy $E$ computed from the predicted states. Left: Models without stability bias; Right: Models with stability bias.

$\boldsymbol{\omega}(t) \in \mathbb{R}^3$ in the principal-axis coordinate frame, the evolution equation is given by

$$\boldsymbol{I}\dot{\boldsymbol{\omega}} + \boldsymbol{\omega} \times (\boldsymbol{I}\boldsymbol{\omega}) = -\mu\boldsymbol{\omega} \quad \forall t > 0. \tag{5}$$

Here, $\boldsymbol{I} = \operatorname{diag}(I_1, I_2, I_3)$ denotes the inertia matrix and $\mu$ is a damping coefficient. By introducing

$$\mathcal{H}(\boldsymbol{\omega}) = \frac{1}{2}I_1\omega_1^2 + \frac{1}{2}I_2\omega_2^2 + \frac{1}{2}I_3\omega_3^2, \tag{6}$$

which describes the kinetic energy $E$, the system can be rewritten in the form of Equation (3), with state-dependent structure matrix $\boldsymbol{J}$, constant dissipation matrix $\boldsymbol{R}$, and vanishing input matrix $\boldsymbol{G}$:

$$\begin{bmatrix} \dot{\omega}_1 \\ \dot{\omega}_2 \\ \dot{\omega}_3 \end{bmatrix} = \left( \underbrace{\begin{bmatrix} 0 & -\frac{I_3}{I_1 I_2}\omega_3 & \frac{I_2}{I_3 I_1}\omega_2 \\ \frac{I_3}{I_1 I_2}\omega_3 & 0 & -\frac{I_1}{I_2 I_3}\omega_1 \\ -\frac{I_2}{I_3 I_1}\omega_2 & \frac{I_1}{I_2 I_3}\omega_1 & 0 \end{bmatrix}}_{\boldsymbol{J}(\boldsymbol{\omega})} - \mu \underbrace{\begin{bmatrix} \frac{1}{I_1^2} & 0 & 0 \\ 0 & \frac{1}{I_2^2} & 0 \\ 0 & 0 & \frac{1}{I_3^2} \end{bmatrix}}_{\boldsymbol{R}} \right) \frac{\partial \mathcal{H}}{\partial \boldsymbol{\omega}}. \tag{7}$$

In anticipation of the numerical experiment in Section 4.1, Figure 1 shows the predicted energy $E$ from various models trained on trajectories from the spinning body system. Those incorporating both, physical biases from the PHS-formulation and stability (bPHNN, sPHNN, and sPHNN-LM) perform best, highlighting the importance of these constraints for learning system dynamics.

## 3 Stable port-Hamiltonian neural networks

Our goal is to learn the dynamics of physical systems in the form of Equation (3) with desirable stability properties from observations. To this end, we parameterize the components $\mathcal{H}$, $\boldsymbol{J}$, $\boldsymbol{R}$ and $\boldsymbol{G}$ of PHS with NNs.

### 3.1 Stability of port-Hamiltonian dynamics

To guarantee the existence of stable equilibria of port-Hamiltonian systems, it is sufficient to enforce the existence of strict local minima in the Hamiltonian. However, these stability implications are only *local*, meaning the size and shape of the corresponding basin of attraction are not constrained by Equation (3). Small perturbations might be enough to stray far from the stable equilibria. Furthermore, the general PHS can represent dynamics such as unbounded trajectories, which must be considered unstable from a practical standpoint. The following describes the necessary requirements for achieving *global* stability. Essentially, the Hamiltonian $\mathcal{H}$ is required to be a suitable Lyapunov function, which includes positive definiteness and radial unboundedness. These properties are achieved by constraining the Hamiltonian to be convex and ensuring the existence of a strict minimum. This approach is similar to the construction of a Lyapunov function by Kolter and Manek [27], but differs in the way the minimum is enforced.

**Theorem 3.1.** *Consider the PHS Equation (3) in the unforced case $\boldsymbol{u}(t) = \boldsymbol{0}$:*

$$\dot{\boldsymbol{x}} = [\boldsymbol{J}(\boldsymbol{x}) - \boldsymbol{R}(\boldsymbol{x})] \frac{\partial \mathcal{H}}{\partial \boldsymbol{x}}(\boldsymbol{x}), \quad \textit{with} \quad \boldsymbol{J} = -\boldsymbol{J}^{\mathsf{T}}, \ \boldsymbol{R} = \boldsymbol{R}^{\mathsf{T}} \succeq 0. \tag{8}$$

*Suppose the Hamiltonian $\mathcal{H}(\boldsymbol{x})$ is convex, twice continuously differentiable, and fulfills:*

$$\mathcal{H}(\boldsymbol{0}) = 0, \qquad \left.\frac{\partial \mathcal{H}}{\partial \boldsymbol{x}}\right|_{\boldsymbol{x}=\boldsymbol{0}} = \boldsymbol{0}, \qquad \left.\frac{\partial^2 \mathcal{H}}{\partial \boldsymbol{x} \partial \boldsymbol{x}}\right|_{\boldsymbol{x}=\boldsymbol{0}} \succ 0. \tag{9}$$

*Then, $\mathcal{H}$ is a suitable Lyapunov function for showing stability of the equilibrium at $\boldsymbol{x}(t) = \boldsymbol{0}$, and all solutions are bounded. Furthermore, the equilibrium is globally asymptotically stable if $\boldsymbol{R}(\boldsymbol{x}) \succ 0$.*

A proof of Theorem 3.1 is provided in Appendix A. However, its implications also follow a clear physical intuition: If a dynamical system continuously loses energy via internal dissipation ($\boldsymbol{R} \succ 0$), it will eventually settle into an energy minimum corresponding to an asymptotically stable equilibrium. The convexity requirement, together with Equation (9), guarantees the existence of a unique strict and global minimum at the origin. Additionally, it ensures that the energy for states at infinity is unbounded, i.e., $\mathcal{H}(\boldsymbol{x}) \to \infty$ as $\|\boldsymbol{x}\| \to \infty$. Thus, no finite state has more energy than a state at infinity, and all trajectories must be bounded. As a result, all solutions eventually converge to the global energy minimum. While convexity is sufficient to ensure global asymptotic stability, it is not a necessary condition. To enhance model expressiveness when needed, techniques such as input warping [27] can be employed, but were not required in the applications considered in this work.

For conservative systems with $\boldsymbol{R} = \boldsymbol{0}$, energy is not dissipated, and attractive equilibria cannot exist. Therefore, a conservative system cannot possess asymptotically stable equilibria. Nevertheless, the minimum in the energy still represents a stable equilibrium, and all trajectories stay bounded. The mixed case, where $\boldsymbol{R} \succeq 0$, only allows the same conclusions as the conservative case. However, unless the dissipation matrix vanishes in a large region in phase space, it is unlikely that a system with $\boldsymbol{R} \succeq 0$ is not globally asymptotically stable.

### 3.2 Neural network model architecture

Having established the theoretical requirements for stability of PHS, we now outline how these conditions can be met with a dedicated NN model. We call the resulting architecture sPHNN. Its computation graph is illustrated in Figure 2a. Figure 2b shows the decomposition of a sPHNN with randomly initialized NNs into conservative and dissipative dynamics, highlighting the model's physical bias and inherent interpretability.

**Hamiltonian** To fulfill the constraints of Theorem 3.1, the Hamiltonian $\mathcal{H}(\boldsymbol{x})$ is represented by a fully input convex neural network (FICNN) [2]. FICNNs can approximate any convex function [8]. They are defined by the recurrence relation

$$
\begin{aligned}
\boldsymbol{z}_1 &= \sigma_0 \left( \boldsymbol{W}_0 \boldsymbol{x} + \boldsymbol{b}_0 \right), \\
\boldsymbol{z}_{i+1} &= \sigma_i \left( \boldsymbol{U}_i \boldsymbol{z}_i + \boldsymbol{W}_i \boldsymbol{x} + \boldsymbol{b}_i \right), \ i = 1, 2, ..., k-1, \\
f(\boldsymbol{x}) &= \boldsymbol{z}_k,
\end{aligned}
\tag{10}
$$

with the pass-through weight matrices $\boldsymbol{W}_i$ that map the input $\boldsymbol{x}$ directly to the activation of the $k-1$ hidden layers, bias vectors $\boldsymbol{b}_i$, and layer outputs $\boldsymbol{z}_i$. If the weight matrices $\boldsymbol{U}_i$ have non-negative

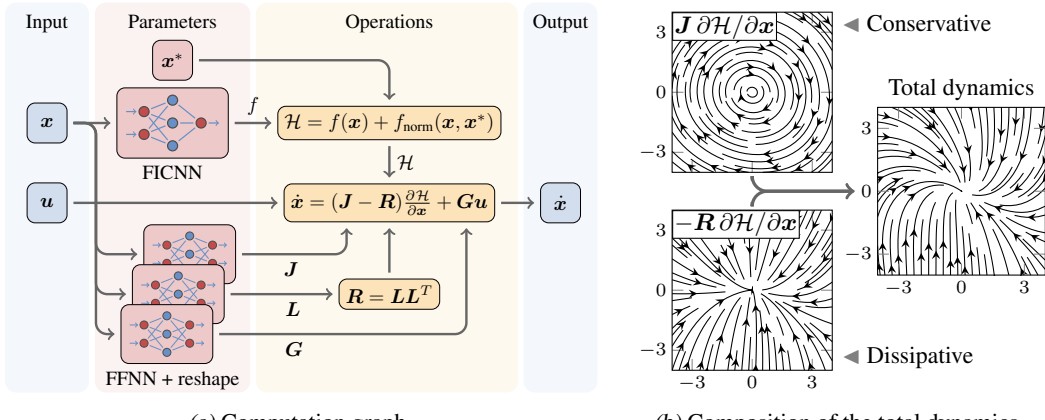

*(a)* Computation graph.      *(b)* Composition of the total dynamics.

*Figure 2. sPHNN model architecture*: (a) Computation graph of sPHNN. The FICNN parameterizing $f$ is normalized to obtain $\mathcal{H}$ and the outputs of the FFNNs for $\boldsymbol{J}$, $\boldsymbol{L}$, and $\boldsymbol{G}$ are reshaped to be skew-symmetric, lower triangular, and rectangular matrices, respectively. (b) Dynamics of a randomly initialized sPHNN for $n = 2$, showing the built-in interpretability through the separation into conservative and dissipative dynamics.

components, the first activation function $\sigma_0$ is convex, and all subsequent $\sigma_i$ are convex and non-decreasing, then the output $f(\boldsymbol{x})$ is convex in the input $\boldsymbol{x}$. Since the Hamiltonian should be twice continuously differentiable, the same must hold for the activation functions. One suitable choice is the softplus activation. The conditions for stability in Equation (9) are then fulfilled by normalizing the FICNN $f$:

$$\mathcal{H}(\boldsymbol{x}) = \underbrace{f(\boldsymbol{x}) - f(\boldsymbol{x}^*) - \frac{\partial f}{\partial \boldsymbol{x}}\bigg|_{\boldsymbol{x}^*}^{\mathsf{T}} (\boldsymbol{x} - \boldsymbol{x}^*)}_{f_{\text{norm}}(\boldsymbol{x}, \boldsymbol{x}^*)} + \underbrace{\epsilon \left\| \boldsymbol{x} - \boldsymbol{x}^* \right\|^2}_{f_{\text{reg}}(\boldsymbol{x}, \boldsymbol{x}^*)} . \tag{11}$$

Consequently, the Hamiltonian $\mathcal{H}$ and its gradient vanish at $\boldsymbol{x}^*$ due to the normalization term $f_{\text{norm}}$. Moreover, the regularization term $f_{\text{reg}}$ with an arbitrarily small $\epsilon > 0$ ensures positive definiteness of the Hamiltonian's Hessian. However, in practice, $f_{\text{reg}}$ can be omitted since the FICNN provides enough bias towards local strict convexity. After training, the stability guarantee can then be recovered by verifying the Hamiltonian's positive definiteness at $\boldsymbol{x}^*$. Choosing $\boldsymbol{x}^* = \boldsymbol{0}$ fulfills all requirements of Theorem 3.1. However, the presented approach is more general. As the assumption that the equilibrium is located at the origin is only made for ease of notation, the normalization approach in Equation (11) can position the stable equilibrium anywhere in phase space and is not restricted to the origin. If the true equilibrium position is known beforehand, fixing $\boldsymbol{x}^*$ introduces an additional inductive bias. While this information can often be obtained from prior physical knowledge, it may not always be available. In such cases, the equilibrium can be inferred directly from data by treating $\boldsymbol{x}^*$ as a trainable parameter during optimization.

**Structure, dissipation and input matrices** The matrix-valued functions $\boldsymbol{J}(\boldsymbol{x})$, $\boldsymbol{R}(\boldsymbol{x})$ and $\boldsymbol{G}(\boldsymbol{x})$ are parameterized by feedforward neural networks (FFNNs), where the outputs are suitably reshaped into matrices. As $\boldsymbol{J}$ is constrained to be skew-symmetric, the respective FFNN's output vector is mapped to the space of skew-symmetric matrices via a linear transformation. Similarly, the positive definiteness (or semi-definiteness) of $\boldsymbol{R}$ is ensured via the Cholesky factorization $\boldsymbol{R} = \boldsymbol{L}\boldsymbol{L}^{\mathsf{T}}$, where the reshaped FFNN output $\boldsymbol{L}$ is a lower triangular matrix with positive (or non-negative) elements along the main diagonal. For some applications, $\boldsymbol{J}$, $\boldsymbol{R}$ and $\boldsymbol{G}$ are independent of $\boldsymbol{x}$, and thus, learning constant matrices is sufficient. In particular, this is the case when the conservative dynamics of the system under consideration are Hamiltonian in the given coordinates. Furthermore, by fixing $\boldsymbol{J}$ to the symplectic matrix (and choosing $\boldsymbol{R} = \boldsymbol{G} = \boldsymbol{0}$), the sPHNN can be reduced to a Hamiltonian NN.

### 3.3 Training stable port-Hamiltonian neural networks

There are two fundamentally different approaches for training time-continuous dynamic models such as NODEs and sPHNNs: *derivative* and *trajectory fitting*. The former directly compares the predicted state derivatives $\dot{\boldsymbol{x}}$ from Equation (3) to the true ones. While this approach is efficient, it requires the true time derivatives to be available as training data, which is typically not the case for real-world measurement data. Furthermore, derivative fitting is not applicable when augmented states are used [12], i.e., when $\boldsymbol{x}$ is padded with additional dimensions compared to the observable states to enrich the representable dynamics. Further, a model proficient in predicting $\dot{\boldsymbol{x}}$ does not necessarily produce accurate predictions for trajectories $\boldsymbol{x}(t)$, as small derivative errors can accumulate exponentially when integrating an ODE [17]. On the other hand, trajectory fitting directly compares integrated model trajectories $\boldsymbol{x}(t)$ with ground-truth trajectories, ensuring long-term accuracy. Here, the optimization process necessitates either the propagation of the loss gradients through the ODE solver or the use of the adjoint sensitivity method [7], which computes the gradients by solving a modified ODE. In both cases, trajectory fitting incurs significant computational costs due to gradient propagation and model integration, making trajectory fitting much slower than derivative fitting.

A distinct advantage of sPHNNs is the favorable influence the stability constraint exerts on the training process when using trajectory fitting. NODEs and similar architectures can suffer from exploding gradients, where small parameter changes lead to large loss gradients. This can slow the training process or prevent it from converging. A potential cause is the crossing between the boundaries of local basins of attraction during training, leading to significant differences in long-term behavior [41]. Since sPHNNs have only one global basin of attraction, this source of training difficulties is eliminated. In our numerical experiments, we observe good convergence of the training error of sPHNNs where NODEs struggle without the use of regularization terms or gradient clipping.

# 4 Experiments

We evaluate the proposed model on multiple example problems, including measurement data from a real physical system and multiphysics simulation data. The performance of sPHNNs is compared to that of NODEs, PHNNs, and bounded PHNNs (bPHNNs). The latter two are PHS-based NN models that do not enforce or relax the convexity constraint of sPHNNs, and thus do not guarantee global stability. In PHNNs, the Hamiltonian is modeled by an unconstrained FFNN, whereas in bPHNNs, it is constructed to be positive and radially unbounded, which ensures bounded trajectories. The latter architecture is intentionally designed to resemble OnsagerNet [55], but with the key difference that it supports arbitrary time-dependent inputs, making it suitable for our experimental setting. For examples that purely rely on derivative fitting, we further include stable NODEs (sNODEs) [27]. Using trajectory fitting, we could not achieve convergence of sNODEs, excluding them from most experiments. This is likely due to difficulties caused by discontinuous dynamics stemming from the projection-based stability enforcement, as also mentioned by [46]. Finally, we include a variant of sPHNNs with learnable minima (sPHNN-LM). In contrast to sPHNNs, which have $\boldsymbol{x}^*$ fixed to an a-priori known equilibrium position, sPHNN-LMs have to infer the equilibrium position from the training data and start out with a randomly initialized $\boldsymbol{x}^*$. All models are trained by minimizing the mean squared error using the ADAM optimizer [25]. Model hyperparameters and computational costs are listed in Appendix B. Predictions are generated by rollout with a numerical integrator using the Runge-Kutta scheme Tsit5, with adaptive step size and a known input function $\boldsymbol{u}(t)$. Code and model weights are available on GitHub[1].

## 4.1 Spinning rigid body

We generate data of a damped spinning rigid body by numerically integrating Euler's rotation equations, which were already introduced in Section 2.2, Equation (5). Here, accurately capturing the dynamics of the angular velocity vector $\boldsymbol{x} = \boldsymbol{\omega}$ with a PHS requires a state-dependent structure matrix $\boldsymbol{J}(\boldsymbol{x})$. We train ten instances per model type using derivative fitting with $(\boldsymbol{\omega}, \dot{\boldsymbol{\omega}})$ pairs sampled from the trajectories. Figure 1 shows the rotational energy $E(\boldsymbol{\omega})$ computed from model predictions $\boldsymbol{\omega}$ via Equation (6). The PHNN, NODE, and sNODE models show deviations from ground truth, with the former two not converging to zero. Furthermore, the NODEs show unphysical behavior with at times increasing energy and exhibit high variance between instances. In contrast, sPHNNs and bPHNNs match the true energy well, highlighting the importance of providing both stability and physical biases. The sPHNN-LM variant recovered the actual equilibrium position accurately, with an interquartile mean distance between the true and inferred minimum across all instances of $0.010$. As a result, its predictive performance is nearly identical to that of the sPHNN with fixed $\boldsymbol{x}^*$.

## 4.2 Cascaded tanks

Next, the sPHNN model is evaluated using the cascaded tanks dataset [47, 48]. It contains measurements of a physical fluid level control system consisting of a pump and two tanks, see Appendix D for details. The training data consists of a single trajectory with 1024 pairs of pump voltages $u$ and water level measurements $y$ of the lower tank. We model the system using a two-dimensional state vector $\boldsymbol{x} = [x_1, x_2]^\intercal$, where $y = x_2$. Assuming the augmented state $x_1$ describes the water level in the upper tank, it is straightforward to determine the equilibrium of the autonomous system. With zero input, the tanks will eventually drain; therefore, the stable equilibrium is given by $\boldsymbol{x}^* = \boldsymbol{0}$. We thus fix the minimum of the sPHNN's Hamiltonian to the origin. To investigate the effects of a potential lack of this knowledge, we also train the sPHNN-LM variant. Its minimum location $\boldsymbol{x}^*$ is randomly initialized and optimized during training. Apart from this, both sPHNN versions are identical and use a constant fixed symplectic matrix as the structure matrix, as well as constant but learnable dissipation and input matrices. The same structure applies to PHNN and bPHNN.

We train 20 instances per model type and show a statistical evaluation of the root mean squared errors (RMSEs) in Figure 3a. Overall, sPHNN, sPHNN-LM, and bPHNN achieve comparable accuracy, while NODE and PHNN perform worse on average across both trajectories. In addition to their better performance, the models with stability bias exhibit lower variance across model instances, indicating more consistent results. To evaluate the models' extrapolation capabilities, we extend the evaluation data with $400\,\mathrm{s}$ of zero-input. The resulting predictions are depicted in

---

[1] `https://github.com/CPShub/sphnn-publication`

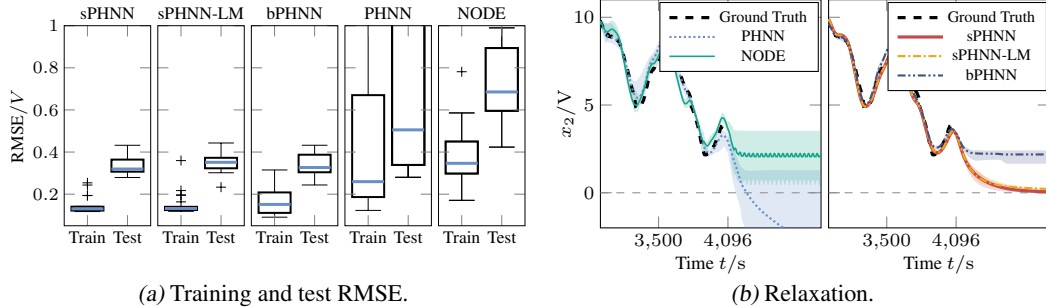

*(a)* Training and test RMSE.                    *(b)* Relaxation.

*Figure 3. Cascaded tanks*: (a) RMSE of the models on training and test trajectory. (b): Predictions for the extended test trajectory. At $t = 4096\,\text{s}$, the pump is turned off. Lines correspond to the interquartile mean and shaded areas represent the interquartile range of the predictions from the 20 model instances.

Figure 3b. Assuming perfectly calibrated sensors, the true system response is an eventual zero output as both tanks drain. By construction, this behavior is guaranteed for sPHNNs. The 20 sPHNN-LM instances reliably identified the systems equilibrium to lie between $-0.196\,\text{V}$ to $0.432\,\text{V}$, which aligns with physical intuition. In contrast the bPHNN, NODE and PHNN models exhibit poor physical extrapolation, predicting equilibria far from zero, diverging water levels, or oscillatory behavior. In an additional experiment presented in Appendix D, the sPHNN models are also shown to have successfully learned the hard saturation effect that occurs when a water tank overflows — a task at which both NODE and PHNN fail.

## 4.3   Thermal food processing surrogate

With the applicability of sPHNNs to real-world data established, this section focuses on one of the promising applications of data-driven dynamic system identification: the efficient construction of surrogate models. Here, data from a conjugate heat transfer simulation of a convection oven coupled with a soft matter model for meats from Kannapinn et al. [21, 23] is employed. The data consists of trajectories representing the temperature histories $T_A$ and $T_B$ at two probe points within the meat. These result from a predefined excitation signal $T_{\text{oven}}$, which controls the oven temperature. The surrogate's task is to predict the probe temperatures given the oven temperature as input.

We fit the models to initially $n_D = 2$ trajectories consisting of 280 samples each (see Appendix E). Per model, we vary the dimensionality of the state $\boldsymbol{x} \in \mathbb{R}^n$ with $n = 2 + n_A$ by adding 0 to 3 augmented dimensions $n_A$. The minimum $\boldsymbol{x}^*$ of the sPHNNs' Hamiltonian is set to the point in phase space corresponding to the ambient temperature, as motivated by physical intuition about the stable thermal equilibrium. The initial values of the augmented states are fixed to zero. In total, 20 instances are trained per model type and augmented dimension.

The RMSEs evaluated on 15 test trajectories (see Appendix E) are depicted in Figure 4a. The sPHNN predictions tend to improve with more augmented dimensions, achieving errors comparable to those obtained by Kannapinn et al. [21] using commercial software. This appears intuitive, as additional hidden state variables may be required to accurately describe the evolution of $T_A$ and $T_B$ that results from the multiphysics, conjugate heat transfer process, see also [12]. However, for bPHNNs and sPHNN-LMs no clear trend emerges. The NODEs and PHNNs even tend to deteriorate in performance with more augmented dimensions, likely due to instabilities in the learned dynamics. This becomes apparent in Figure 4c (top row), which depicts the models' predictions for an extended test case. While the sPHNNs consistently converge to temperatures close to thermal equilibrium, the predictions of NODEs and PHNNs tend to become unstable when one or more augmented dimensions are used. Although all models improve with more training data (Figure 4b), these instabilities persist when extrapolating in time (Figure 4c, bottom row $t > 1395\,\text{s}$). In contrast, the stability constraint of the sPHNNs and sPHNN-LMs allows safe use of the additional flexibility provided by a higher-dimensional state space, avoiding the pitfalls observed in the unconstrained models. Given sufficient training data, the sPHNN-LM variant performs on par with the sPHNN, whereas in the scarce-data regime its errors increase and are comparable to those of the bPHNNs. Although the bPHNNs also remain stable, they are consistently outperformed by the sPHNNs for $n_A = 3$, regardless of dataset size, and especially when only a single training trajectory is available $n_D = 1$, see Figure 4b and Figure 4c (bottom left).

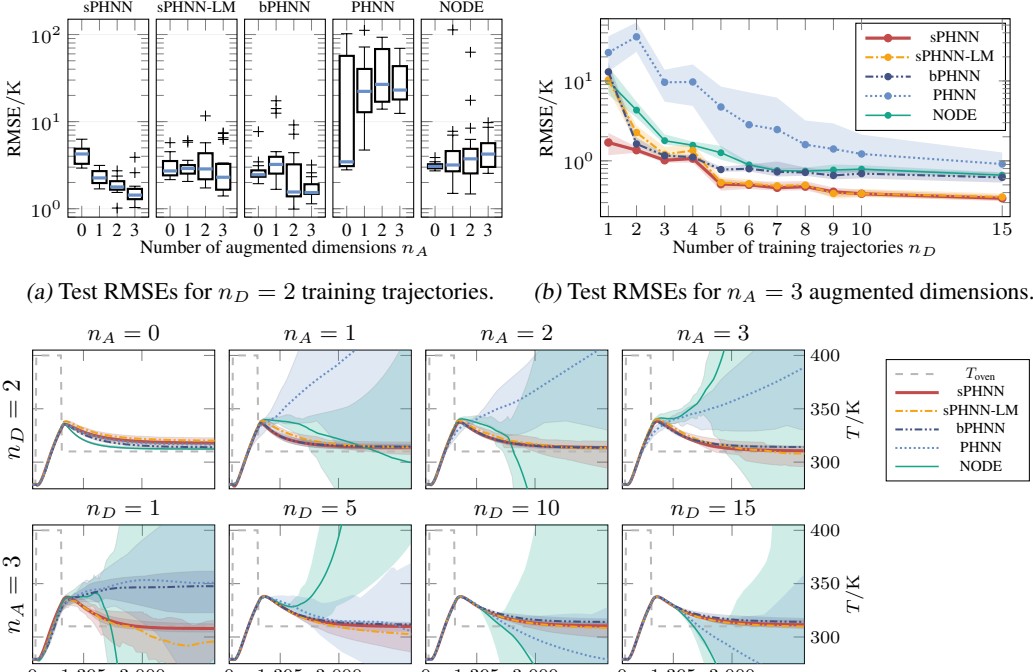

*(a)* Test RMSEs for $n_D = 2$ training trajectories.

*(b)* Test RMSEs for $n_A = 3$ augmented dimensions.

*(c)* Model predictions for different combinations of augmented dimensions and training data.

*Figure 4. Thermal food processing surrogate*: (a) and (b) RMSEs evaluated on 15 test trajectories for various numbers of augmented dimensions $n_A$ and training trajectories $n_D$. (c) Interquartile mean and range of the $T_A$ predictions for a custom test case. *Top row*: Varying number of augmented dimensions with fixed number $n_D = 2$ of training trajectories. *Bottom row*: Varying number of training trajectories with fixed number $n_A = 3$ of augmented dimensions. The time $t = 1395\,\text{s}$ marks the length of the training trajectories.

*Table 1.* Median RMSE of 20 instances per models using $n_A = 3$ augmented dimensions, trained on $n_D = 2$ noisy trajectories and evaluated on noise-free training and test sets.

| Model | Training | | | Test | | |
|---|---|---|---|---|---|---|
| | 5 % noise | 15 % noise | 25 % noise | 5 % noise | 15 % noise | 25 % noise |
| sPHNN | **0.332** | **0.854** | **1.354** | **1.170** | **2.595** | **3.177** |
| sPHNN-LM | 0.638 | 1.096 | 1.622 | 2.420 | 2.888 | 3.677 |
| bPHNN | 0.500 | 1.049 | 2.207 | 1.572 | 2.828 | 5.231 |
| PHNN | 2.828 | 6.750 | 5.779 | 28.233 | 25.631 | 24.224 |
| NODE | 0.993 | 1.974 | 2.423 | 3.577 | 3.482 | 4.362 |

To assess the models' robustness to noise, we conducted an additional experiment using $n_A = 3$ augmented dimensions and $n_D = 2$ training trajectories, introducing zero-mean Gaussian noise for both the input $T_{\text{oven}}$ and the outputs $T_A$ and $T_B$. The noise amplitudes were varied between 5 % and 25 % of the original signals' standard deviations. Table 1 reports the average RMSE over 20 model instances per model type, evaluated on noise-free test data. To isolate the effect of noise and eliminate generalization error, we also evaluated the RMSE on the noise-free training data. All models demonstrate a degree of robustness to noise, since the integration step involved in trajectory fitting acts as a low-pass filter, effectively smoothing the learned dynamics. As a result, high-frequency Gaussian noise is attenuated, which helps prevent overfitting and improves generalization. Nevertheless, the sPHNN consistently outperforms the baselines on both the training and test sets.

## 4.4 Additive manufacturing surrogate

Finally, we evaluate the applicability of sPHNNs to higher-dimensional state spaces and build a reduced order model (ROM) as a surrogate of a 3D partial differential equation problem. To this end, we generate field data by numerically solving the heat conduction equation for a moving heat

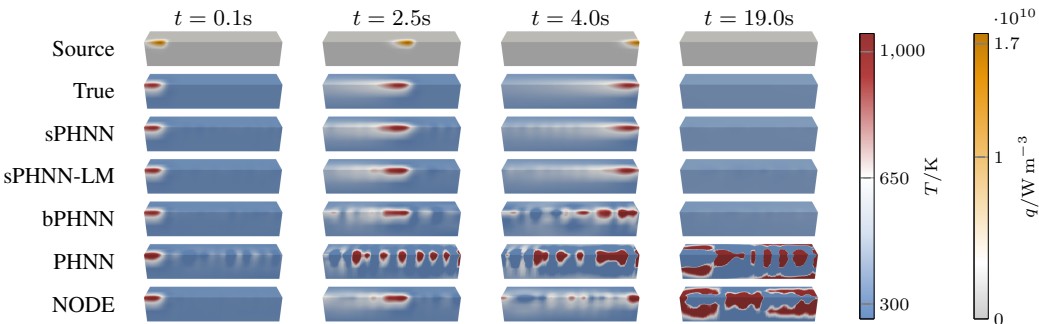

*Figure 5.* Heat source field and temperature field predictions on the cuboid domain for a test case with $v = 12.5\,\text{mm}\,\text{s}^{-1}$, $Q = 400\,\text{W}$. The instances selected for this evaluation resulted in the median test error for the corresponding model type. Colors are clipped to remain in the legend's range.

source on a cuboid domain with convective and radiative thermal boundary conditions. The finite element simulation is designed to model the evolution of the temperature field of a metal additive manufacturing process, see [22] for details. In total, 25 trajectories are obtained by varying the heat source speed $v$ from $10\,\text{mm}\,\text{s}^{-1}$ to $20\,\text{mm}\,\text{s}^{-1}$ and power $Q$ from $300\,\text{W}$ to $500\,\text{W}$. All trajectories span $20\,\text{s}$, with a non-zero heat source lasting for the initial $2.5\,\text{s}$ to $6\,\text{s}$, depending on $v$. However, directly using the discretized field data would lead to very high-dimensional models and render the training and inference inefficient. We thus apply a proper orthogonal decomposition (POD) to map the temperature and heat source fields onto $40$-dimensional latent spaces, respectively (see Appendix F for details). The models are trained on the latent representations of the two trajectories with $(v, Q) = (10\,\text{mm}\,\text{s}^{-1}, 300\,\text{W})$ and $(20\,\text{mm}\,\text{s}^{-1}, 500\,\text{W})$ using derivative fitting and subsequent fine-tuning with trajectory fitting.

Figure 5 depicts the heat source field (top row) and true and predicted temperature fields from the different model types. While the sPHNN's total test RMSE peaks below $30\,\text{K}$ (see Appendix F, Figure F.2) and decreases as the system approaches thermal equilibrium, the errors for NODE and PHNN increase rapidly, driven by instabilities in the learned dynamics. These instabilities are likely caused by the sparsity of the training data, which leaves gaps in the phase space, allowing unstable dynamics to emerge. The stability constraint of sPHNN and sPHNN-LM mitigates these issues, ensuring robust performance even with scarce training data, while outperforming the bPHNN thanks to their stronger inductive bias.

## 5 Conclusion

This work proposes stable port-Hamiltonian neural networks (sPHNNs) for accurate, robust, and reliable identification of stable nonlinear dynamics. They leverage the port-Hamiltonian framework, which provides a basis for model interpretation and ensures adherence to the laws of thermodynamics in terms of energy conservation or dissipation balances with respect to a learned Hamiltonian energy function. Additionally, the approach ensures global asymptotic stability of the identified dynamics, without requiring projection, by constraining the Hamiltonian to be a convex, positive definite Lyapunov function. While this limits sPHNNs to modeling systems that appear globally stable within the region of interest, it provides a strong inductive bias that enables robust learning and reasonable generalization from sparse data. We demonstrate the approach's viability using data from synthetic examples, real-world measurements, and complex multiphysics simulation models. The resulting models, built with small neural network architectures, exhibit low variance across instances, accurate prediction, and robust generalization. Furthermore, the proposed method can successfully exploit augmented dimensions where unconstrained alternative models struggle with instabilities or data scarcity. While the approach can identify the equilibrium point from data, we find that providing the equilibrium as a prior can further enhance prediction accuracy and reduce data requirements. Future work will extend the approach to multistable systems by allowing Hamiltonians with multiple minima, while retaining favorable stability properties.

## Acknowledgments and Disclosure of Funding

The authors acknowledge the financial support provided by the Deutsche Forschungsgemeinschaft (DFG, German Research Foundation, research grant number 492770117 and the CRC/TRR 361 CREATOR), the Hessian Center for Artificial Intelligence (Hessian.AI, project EnDyRo), and the Graduate School Computational Engineering at TU Darmstadt. Furthermore, we would like to thank An Thai Le, Daniel Palenicek, and Joe Watson from TU Darmstadt (IAS) for their constructive feedback on the manuscript.

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

# A Proof of stability

This section provides a proof of Theorem 3.1, which describes the essential requirements for achieving stable dynamics with port-Hamiltonian systems. Due to the relevance of convexity to the following discussion, the following Lemma first provides some widely known equivalent formulations of convexity for scalar-valued multivariate functions.

**Lemma A.1.** *Consider a twice continuously differentiable function $f : \mathbb{R}^n \to \mathbb{R}$, $\boldsymbol{x} \mapsto f(\boldsymbol{x})$. If $f$ is convex, the following statements hold:*

1. $\lambda f(\boldsymbol{x}_1) + (1 - \lambda)f(\boldsymbol{x}_2) \geq f(\lambda \boldsymbol{x}_1 + (1 - \lambda)\boldsymbol{x}_2) \quad \forall \boldsymbol{x}_1, \boldsymbol{x}_2 \in \mathbb{R}^n, \ \lambda \in [0, 1]$,

2. $\frac{\partial f}{\partial \boldsymbol{x}} \big|_{\boldsymbol{x}_1}^\mathsf{T} (\boldsymbol{x}_2 - \boldsymbol{x}_1) \leq f(\boldsymbol{x}_2) - f(\boldsymbol{x}_1) \quad \forall \boldsymbol{x}_1, \boldsymbol{x}_2 \in \mathbb{R}^n$,

3. $\frac{\partial^2 f}{\partial \boldsymbol{x} \partial \boldsymbol{x}}(\boldsymbol{x}) \succeq 0 \quad \forall \boldsymbol{x} \in \mathbb{R}^n$.

For a proof, the reader is referred to [6]. Note that for strict convexity, the inequalities in statements 1 and 2 must be strict for all $\boldsymbol{x}_1 \neq \boldsymbol{x}_2$, and the Hessian in statement 3 must be positive definite. With the notion of convexity established, attention can now be directed to the proof of Theorem 3.1. For convenience, the statement of the theorem is repeated verbatim below:

**Theorem A.2.** *Consider the PHS Equation (3) in the unforced case $\boldsymbol{u}(t) = \boldsymbol{0}$:*

$$\dot{\boldsymbol{x}} = [\boldsymbol{J}(\boldsymbol{x}) - \boldsymbol{R}(\boldsymbol{x})] \frac{\partial \mathcal{H}}{\partial \boldsymbol{x}}(\boldsymbol{x}), \quad \text{with} \quad \boldsymbol{J} = -\boldsymbol{J}^\mathsf{T}, \ \boldsymbol{R} = \boldsymbol{R}^\mathsf{T} \succeq 0. \tag{A.1}$$

*Suppose the Hamiltonian $\mathcal{H}(\boldsymbol{x})$ is convex, twice continuously differentiable, and fulfills:*

$$\mathcal{H}(\boldsymbol{0}) = 0, \qquad \frac{\partial \mathcal{H}}{\partial \boldsymbol{x}} \bigg|_{\boldsymbol{x}=\boldsymbol{0}} = \boldsymbol{0}, \qquad \frac{\partial^2 \mathcal{H}}{\partial \boldsymbol{x} \partial \boldsymbol{x}} \bigg|_{\boldsymbol{x}=\boldsymbol{0}} \succ 0. \tag{A.2}$$

*Then, the system in Equation (A.1) has a stable equilibrium at $\boldsymbol{x}(t) = \boldsymbol{0}$, and all solutions are bounded. Furthermore, the equilibrium is globally asymptotically stable if $\boldsymbol{R}(\boldsymbol{x}) \succ 0$.*

*Proof.* Since the gradient of the Hamiltonian vanishes and its Hessian is positive definite at $\boldsymbol{x} = \boldsymbol{0}$, it has a strict local minimum at the origin. This implies the existence of an $R > 0$ such that

$$\mathcal{H}(\boldsymbol{x}) > \mathcal{H}(\boldsymbol{0}) = 0 \quad \forall \boldsymbol{x} \in \{\|\boldsymbol{x}\| \leq R, \ \boldsymbol{x} \neq \boldsymbol{0}\}. \tag{A.3}$$

Consider the function $g(\boldsymbol{x}) = \frac{a}{r} \|\boldsymbol{x}\|$ for some $r \in (0, R]$ with $a = \inf_{\|\boldsymbol{x}\|=r} \mathcal{H}(\boldsymbol{x}) > 0$. We claim that $g$ is a lower bound for $\mathcal{H}$, i.e. $g(\boldsymbol{x}) \leq \mathcal{H}(\boldsymbol{x})$ for all $\|\boldsymbol{x}\| > r$.

Suppose there existed an $\tilde{\boldsymbol{x}} \in \mathbb{R}^n$ that violated the claim, that is $\|\tilde{\boldsymbol{x}}\| > r$ and $g(\tilde{\boldsymbol{x}}) > \mathcal{H}(\tilde{\boldsymbol{x}})$. Due to the convexity of $\mathcal{H}$, it holds

$$\lambda \mathcal{H}(\tilde{\boldsymbol{x}}) + (1 - \lambda)\mathcal{H}(\boldsymbol{0}) \geq \mathcal{H}(\lambda \tilde{\boldsymbol{x}} + (1 - \lambda) \cdot \boldsymbol{0}) \quad \forall \lambda \in [0, 1] \tag{A.4}$$

and thus $\lambda g(\tilde{\boldsymbol{x}}) > \lambda \mathcal{H}(\tilde{\boldsymbol{x}}) \geq \mathcal{H}(\lambda \tilde{\boldsymbol{x}})$. Choosing $\lambda = \frac{r}{\|\tilde{\boldsymbol{x}}\|}$ and applying the definition of $g$ we obtain

$$\frac{r}{\|\tilde{\boldsymbol{x}}\|} g(\tilde{\boldsymbol{x}}) = a > \mathcal{H}\left(\frac{r}{\|\tilde{\boldsymbol{x}}\|}\tilde{\boldsymbol{x}}\right) \geq a. \tag{A.5}$$

This is a contradiction, and therefore $g$ is a lower bound.

With the bound established, showing that $\mathcal{H}$ is radially unbounded is straightforward. For any path $\boldsymbol{x}(t)$ with $\lim_{t\to\infty} \|\boldsymbol{x}(t)\| = \infty$, once $\|\boldsymbol{x}(t)\| > r$ we have

$$\lim_{t\to\infty} \mathcal{H}(\boldsymbol{x}(t)) \geq \lim_{t\to\infty} g(\boldsymbol{x}(t)) = \infty. \tag{A.6}$$

We would like to remark that the same conclusion could alternatively be reached by showing that $\mathcal{H}$ has at least one bounded sublevel set due to its strict minimum and then concluding radial unboundedness (i.e., coercivity) using Corollary 8.7.1 in [45] together with the standard equivalence for proper closed convex functions. However, the presented approach using a lower bound can directly be extended to show global positive definiteness of $\mathcal{H}$. For this, note that for all $\|\boldsymbol{x}\| > r$ the lower bound implies $0 < g(\boldsymbol{x}) \leq \mathcal{H}(\boldsymbol{x})$. Together with Equation (A.3), this implies that $\mathcal{H}$ is indeed positive definite for all $\boldsymbol{x} \in \mathbb{R}^n$.

From the port-Hamiltonian structure of Equation (A.1) we immediately obtain the decrease condition $\dot{\mathcal{H}} \leq 0$ (see Equation (4)). Using the $\mathcal{H}$ as a Lyapunov function, we obtain all requirements to conclude *local* stability of the equilibrium. Furthermore, the decrease condition implies that a solution starting at $\boldsymbol{x}^*$ cannot leave the sublevel set $\Omega_{\mathcal{H}} = \{\boldsymbol{x} \in \mathbb{R}^n | \mathcal{H}(\boldsymbol{x}) \leq \mathcal{H}(\boldsymbol{x}^*)\}$. Since $\Omega_{\mathcal{H}} \subseteq \Omega_g = \{\boldsymbol{x} \in \mathbb{R}^n | g(\boldsymbol{x}) \leq \mathcal{H}(\boldsymbol{x}^*)\}$ and $\Omega_g$ is clearly bounded, all solutions of Equation (A.1) must be bounded as well.

Finally, to show that *global* asymptotic stability follows from $\boldsymbol{R} \succ 0$, it remains to be demonstrated that $\dot{\mathcal{H}} < 0$ for all $\boldsymbol{x} \neq \boldsymbol{0}$. This is implied by Equation (4) if $\frac{\partial \mathcal{H}}{\partial \boldsymbol{x}} \neq \boldsymbol{0}$ for all $\boldsymbol{x} \neq \boldsymbol{0}$. Proving that the gradient of $\mathcal{H}$ vanishes only at the origin is done by contradiction. Assume that there exists an $\tilde{\boldsymbol{x}} \neq \boldsymbol{0}$ such that $\frac{\partial \mathcal{H}}{\partial \boldsymbol{x}}\big|_{\tilde{\boldsymbol{x}}} = \boldsymbol{0}$. Furthermore, let $\boldsymbol{x}^* = \boldsymbol{0}$. Since $\mathcal{H}$ is convex it follows (see Lemma A.1) that

$$\mathcal{H}(\boldsymbol{x}^*) - \mathcal{H}(\tilde{\boldsymbol{x}}) \geq \left.\frac{\partial \mathcal{H}}{\partial \boldsymbol{x}}\right|_{\tilde{\boldsymbol{x}}}^{\mathsf{T}} (\boldsymbol{x}^* - \tilde{\boldsymbol{x}}) = 0, \tag{A.7}$$

$$\mathcal{H}(\tilde{\boldsymbol{x}}) - \mathcal{H}(\boldsymbol{x}^*) \geq \left.\frac{\partial \mathcal{H}}{\partial \boldsymbol{x}}\right|_{\boldsymbol{x}^*}^{\mathsf{T}} (\tilde{\boldsymbol{x}} - \boldsymbol{x}^*) = 0. \tag{A.8}$$

This implies $\mathcal{H}(\boldsymbol{x}^*) = \mathcal{H}(\tilde{\boldsymbol{x}}) = 0$. However, since $\mathcal{H}$ is globally positive definite and thus only vanishes at the origin, we have $\boldsymbol{x}^* = \tilde{\boldsymbol{x}}$. This is a contradiction as $\tilde{\boldsymbol{x}} \neq \boldsymbol{0}$, which concludes the proof. $\square$

# B  Experiment hyperparameters

In each experiment presented within this manuscript, the different models were trained for the same number of steps using identical learning rates. The mean squared error loss function and the ADAM [25] optimizer were used throughout. The model and training hyperparameters, along with the number of model instances trained for statistical evaluation, are provided in Table 2. Throughout the experiments presented in Sections 4.1 to 4.3, all NNs consisted of two hidden layers, each containing 16 neurons. The network width was increased to 32 neurons for the additive manufacturing surrogate in Section 4.4. All network weights were initialized using the Glorot uniform [15] scheme.

The PHS-based models (sPHNN, sPHNN-LM, PHNN, bPHNN) share the same structure, differing only in how the Hamiltonian $\mathcal{H}$ is parametrized. Specifically, the sPHNNs use an FICNN with a normalization term, as described in Section 3. The only difference for the sPHNN-LM variant is that the minimum $\boldsymbol{x}^*$ of the Hamiltonian is learned as an additional parameter. The PHNNs, by contrast, use an unconstrained FFNN without any normalization. Finally, the bPHNNs ensures positivity and radially unboundedness of the Hamiltonian by adopting the method described by Yu et al. [55] for the OnsagerNets Energy function. Here, the output of an unconstrained network is squared and combined with a quadratic regularization term. However, unlike OnsagerNet, we used standard FFNNs instead of ResNets.

The models were trained on a Windows machine with Intel i7-13700K (13th Generation) and 32 GB of memory. Table 3 provides representative computational costs for training and prediction. Here, *Derivative evaluation* refers to the average time taken by each model to compute the state derivative $\dot{\boldsymbol{x}}$

*Table 2.* Hyperparameters per experiment. TF and DF refer to trajectory and derivative fitting, respectively. The number of instances trained per model type is listed in the "# inst." column, and $n$ and $m$ denote the number of dimensions of state and input. The final three columns list the parametrization of the matrices for sPHNN, sPHNN-LM, bPHNN, and PHNN as either state-dependent matrices (s.) or constant matrices (c.).

| Experiment | # training steps | learning rate | # inst. | $n$ | $m$ | $\boldsymbol{J}$ | $\boldsymbol{R}$ | $\boldsymbol{G}$ |
|---|---|---|---|---|---|---|---|---|
| Randomly initialized model | - | - | 1 | 2 | - | s. | s. | - |
| Spinning rigid body | 50 000 DF | $1 \times 10^{-3}$ | 10 | 3 | - | s. | c. | - |
| Cascaded tanks | 50 000 TF | $5 \times 10^{-4}$ | 20 | 2 | 1 | c. | c. | c. |
| Food processing surrogate | 30 000 TF | $1 \times 10^{-4}$ | 20 | 2 to 5 | 1 | c. | c. | c. |
| Additive manufacturing surrogate | 20 000 DF and 10 000 TF | $1 \times 10^{-3}$ $1 \times 10^{-5}$ | 20 | 40 | 40 | c. | c. | c. |

*Table 3.* Representative computational costs for training and prediction. The top two rows report the training times for derivative and trajectory fitting, respectively. The bottom two rows show the time for a single derivative evaluation and a full trajectory integration using data and models from Section 4.3.

|  | sPHNN | sPHNN-LM | sNODE | bPHNN | PHNN | NODE |
|---|---|---|---|---|---|---|
| Derivative fit Section 4.1 (min) | 0.577 | 0.644 | 0.672 | 0.635 | 0.539 | 0.346 |
| Trajectory fit Section 4.3 (min) | 37.663 | 39.160 | - | 38.329 | 19.111 | 17.951 |
| Derivative evaluation (ms) | 0.007 | 0.007 | - | 0.007 | 0.007 | 0.005 |
| Model integration (ms) | 0.365 | 0.393 | - | 0.422 | 0.237 | 0.221 |

for a given state $x$ and input $u$, which needs to be evaluated for time integration. In contrast, *Model integration* denotes the time required to generate a single trajectory prediction via integration. While the latter better reflects practical use cases, the reported integration time also depends on the integration scheme and chosen step size. Since we use an adaptive step size controller, the step size in turn depends on the learned dynamics.

## C  Spinning rigid body system

The governing equations of the three-dimensional spinning rigid body used in the experiment in Section 4.1 are described in Section 2.2. We generated $(\omega, \dot{\omega})$ pairs for training by evaluating Equation (5) for varying $\omega$. To simulate a measurement process, we selected $\omega$ values from $50\,\mathrm{s}$ long numerically integrated trajectories, using 10 randomly chosen initial conditions $\omega(0) \in [0, 1]^3$ with parameters $I = \mathrm{diag}(1, 2, 3)$ and $\mu = 0.01$. A representative trajectory of $\omega$ is shown in Figure C.1.

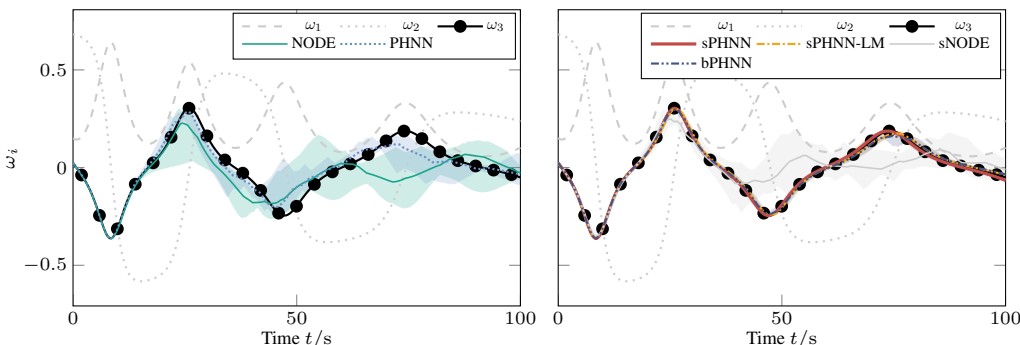

*Figure C.1.* Representative, extended training trajectory for spinning rigid body, including interquartile mean and range of model predictions for $\omega_3$. Left: Models without stability bias; Right: Models with stability bias.

## D  Cascaded tanks data

The cascaded tanks dataset [47, 48] used in Section 4.2 consists of measurements of a fluid level control system. Liquid flows from an upper tank through an outlet into a lower tank, exits via a similar outlet into a reservoir, and is pumped back to the upper tank, see Figure D.1a. The system's input $u$ is the voltage applied to the pump, with higher values resulting in a faster volume flow. When the pump speed is large enough, an overflow can occur in the upper tank. A capacitive sensor measures the fluid level in the lower tank, and the resulting voltage represents the system's output $y$. The dataset consists of two trajectories comprising 1024 pairs $(u, y)$ each and Figure D.1b depicts the training trajectory. The initial water level in the upper tank is unknown, but it is the same for both trajectories.

Besides the experiments presented in Section 4.2, we further investigated the models' capabilities to learn the hard saturation that the cascaded tanks data system exhibits at an output value of $10\,\mathrm{V}$. This occurs when the tanks overflow in response to a large input voltage over an extended period of time (see Figure D.1b) [48]. We explore the models' behavior in this regime by constructing a pump voltage signal that linearly increases from $1\,\mathrm{V}$ initially to $8\,\mathrm{V}$ at $2800\,\mathrm{s}$. The resulting predictions

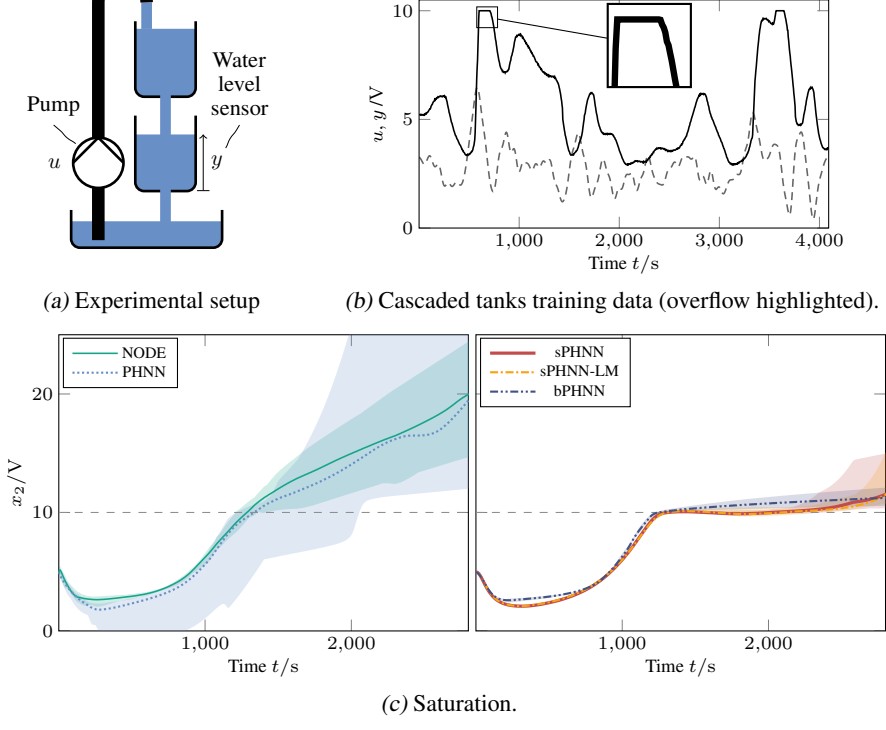

*(a)* Experimental setup    *(b)* Cascaded tanks training data (overflow highlighted).

*(c)* Saturation.

*Figure D.1. Cascaded tanks*: (a) System setup, (b) Training trajectory. (c) Predictions for a linearly increasing pump voltage. Due to an overflow, the true system exhibits a hard saturation at $x_2 = 10$ V. Lines correspond to the interquartile mean, whereas shaded areas represent the interquartile range of the predictions from the 20 model instances.

are shown in Figure D.1c. For pump speeds up to $\sim$7 V ($\sim$2400 s), the sPHNN, sPHNN-LM, and bPHNN instances capture the saturation effect, while the NODE and PHNN instances fail to do so.

## E    Thermal food processing surrogate data

This section provides additional details about the data used in Section 4.3, which is a subset of the data generated and used by Kannapinn et al. [21] in the creation of ROMs. The dataset describes the thermal processing of chicken breasts in a convection oven and was generated via the simulation of a soft-matter model for meats implemented in COMSOL Multiphysics. Each trajectory in the dataset describes the temperature history at two points inside the full order model. These result from a predefined excitation signal that controls the oven temperature. The probe points are situated in the center and close to the surface of the food item, with the corresponding temperatures labeled as $T_A$ and $T_B$, respectively. To facilitate cross-referencing, the alphanumeric identifiers used by Kannapinn et al. [21] for the individual trajectories are retained. The training data consists of trajectories 745 and 795. They use an amplitude-modulated pseudo-random binary sequence (APRBS) and a multi-sine as the excitation signal $T_{\text{oven}}$. Figure E.1 (left and middle) illustrates $T_{\text{oven}}$ along with the core and surface temperatures for both trajectories. This selection provides a fair evaluation of the models across the relevant temperature spectrum. For studying the impact of the number of training trajectories on the models, we additionally included 13 trajectories with identifiers 835, 851, 853, 861, 868, 873, 874, 888, 899, 902, 905, 906 and 917. In the experiment, the test group AP15 [21] is utilized as test data. It comprises 15 trajectories with APRBS excitations, specifically selected to ensure an even distribution of amplitudes and median values. Figure E.1 (right) shows an exemplary test trajectory, including the predictions for $T_A$ from all models using 3 augmented dimensions. All trajectories span 1395 s and consist of 280 samples. The data is normalized for training to ensure the numerical values are of magnitude one. For this, an affine transformation is applied to $T_{\text{oven}}$, $T_A$, and $T_B$ individually. It shifts the ambient temperature to a value of zero and rescales the signal to have unit variance. The scaling and shifting factors are calculated using only the training data to ensure that the test data does not influence the training procedure.

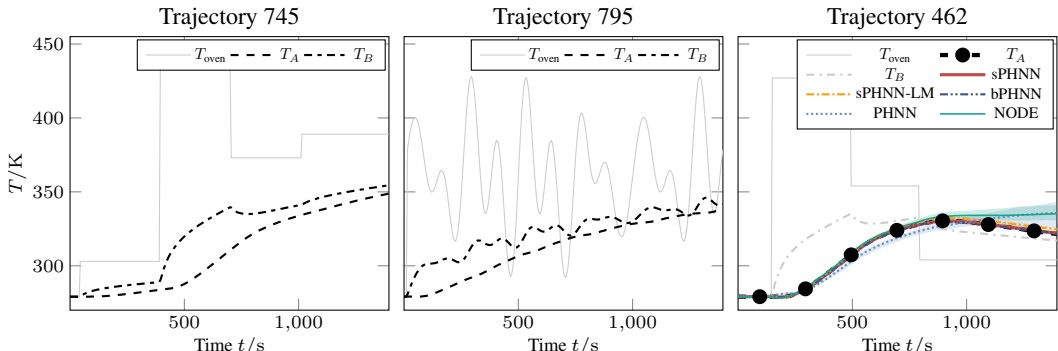

*Figure E.1. Left and middle*: Training trajectories. *Right*: Exemplary test trajectory with interquartile mean and range of the model predictions for $T_A$.

# F  Thermal field data

## F.1  Dimensionality reduction

The following describes the POD method for dimensionality reduction of the field data applied in Section 4.4. The process is first described for general field data given as a matrix $\boldsymbol{A} = [\boldsymbol{a}_0, \ldots, \boldsymbol{a}_M]^\mathsf{T}$ of $M$ snapshots $\boldsymbol{a}_i \in \mathbb{R}^N$, where $M$ is the number of snapshots and $N$ is the number of finite element nodes. To identify the dominant modes of the data, we perform a singular value decomposition (SVD) of $\boldsymbol{A}$:

$$\boldsymbol{A} = \boldsymbol{U}\boldsymbol{\Sigma}\boldsymbol{V}^\mathsf{T} \in \mathbb{R}^{M \times N}, \tag{F.1}$$

where $\boldsymbol{U} \in \mathbb{R}^{M \times M}$ and $\boldsymbol{V} \in \mathbb{R}^{N \times N}$ are unitary matrices and $\boldsymbol{\Sigma} \in \mathbb{R}^{M \times N}$ is a rectangular diagonal matrix containing the singular values $\sigma_1 \geq \cdots \geq \sigma_M$ on its diagonal. Dimensionality reduction is achieved by truncating the SVD $\boldsymbol{A} \approx \tilde{\boldsymbol{U}}\tilde{\boldsymbol{\Sigma}}\tilde{\boldsymbol{V}}^\mathsf{T}$, where $\tilde{\boldsymbol{U}} \in \mathbb{R}^{M \times n}$ and $\tilde{\boldsymbol{V}} \in \mathbb{R}^{N \times n}$ contain the first $n$ columns of $\boldsymbol{U}$ and $\boldsymbol{V}$, and $\tilde{\boldsymbol{\Sigma}} = \mathrm{diag}(\sigma_1, \ldots, \sigma_n)$ is the truncated matrix of the dominant singular values. The dominant modes in the data are then given by $\boldsymbol{M} = c\tilde{\boldsymbol{V}}$. Here, the scaling factor $c$ is the standard deviation of the elements in $\tilde{\boldsymbol{U}}\tilde{\boldsymbol{\Sigma}}$. This scaling ensures that the latent trajectories will have unit variance. To map a latent state $\boldsymbol{x} \in \mathbb{R}^n$ to the full-order field $\boldsymbol{X} \in \mathbb{R}^N$ we compute:

$$\boldsymbol{X} = \boldsymbol{M}\boldsymbol{x}. \tag{F.2}$$

Conversely, to map from the full-order field to the latent space, we solve the least-squares regression problem:

$$\boldsymbol{x} = \arg\min_{\boldsymbol{x}'} \|\boldsymbol{X} - \boldsymbol{M}\boldsymbol{x}'\| = \frac{1}{c^2}\boldsymbol{M}^\mathsf{T}\boldsymbol{X}. \tag{F.3}$$

In Section 4.4, we consider two fields: The thermal field describing the state of the system and the excitation given by the heat source field. For each, a separate SVD is performed to compute $\boldsymbol{M}_T$ and $\boldsymbol{M}_q$ using the respective snapshots from the two training trajectories. We thus obtain two pairs of encoders $\psi$ and decoders $\phi$:

$$\psi_T(\boldsymbol{T}) = \arg\min_{\boldsymbol{x}} \|\boldsymbol{T} - \boldsymbol{M}_T\boldsymbol{x}\|, \qquad\qquad \phi_T(\boldsymbol{x}) = \boldsymbol{M}_T\boldsymbol{x}, \tag{F.4}$$

$$\psi_q(\boldsymbol{q}) = \arg\min_{\boldsymbol{u}} \|\boldsymbol{q} - \boldsymbol{M}_q\boldsymbol{u}\|, \qquad\qquad \phi_q(\boldsymbol{u}) = \boldsymbol{M}_q\boldsymbol{u}. \tag{F.5}$$

To fully leverage the benefits of sPHNNs, it is essential to ensure that the absence of a heat source is encoded as the zero-vector in the corresponding latent space. Since the mapping from the latent space Equation (F.2) is linear, and $\boldsymbol{q} = \boldsymbol{0}$ corresponds to the absence of a heat source in the full-order space, this requirement is automatically fulfilled. However, for applications where a nonzero state $\tilde{\boldsymbol{x}}$ describes the absence of an external energy source or for general nonlinear latent mappings (such as autoencoders), the condition can be met by applying a constant translation in the latent space, which shifts the encoded state $\tilde{\boldsymbol{x}} = \psi(\tilde{\boldsymbol{x}})$ to the origin. In Section 4.4, we use this method to ensure that the thermal equilibrium corresponds to the origin of the temperature latent space. While this is not strictly necessary, knowledge of the equilibrium location in the latent space enables its integration

into the sPHNN's architecture by fixing the stable equilibrium position. Consequently, we arrive at the representation of the latent state

$$\boldsymbol{x} = \psi_T(\boldsymbol{T}) - \psi_T(\boldsymbol{T}_{\text{eq}}), \qquad\qquad \boldsymbol{T} \approx \phi_T(\boldsymbol{x} + \psi_T(\boldsymbol{T}_{\text{eq}})). \qquad\qquad \text{(F.6)}$$

where $\boldsymbol{T}_{\text{eq}}$ is the full-order state vector describing thermal equilibrium. The complete model architecture is represented in Figure F.1. There, the function $\boldsymbol{f}$ is used as a placeholder for an arbitrary evolution equation, to generalize to all model architectures used in Section 4.4.

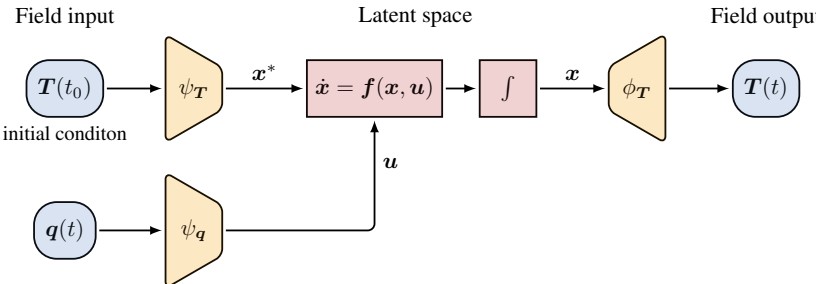

*Figure F.1.* Model setup used in Section 4.4. The initial temperature field $\boldsymbol{T}(t_0)$ and input field $\boldsymbol{q}(t)$ are mapped via encoders $\phi_T, \phi_q$ into the latent space, where the latent dynamics $\boldsymbol{f}$ are integrated. Here, $\boldsymbol{f}$ is used as a placeholder for a specific model, e.g., NODE or sPHNN. Finally, the predicted latent trajectory $\boldsymbol{x}(t)$ is mapped to the full-order field via the decoder $\phi_T$.

Finally, we note that alternative dimensionality reduction techniques could be applied. In preliminary experiments, we explored the use of autoencoders for this purpose. Although autoencoders offer advantages over the linear SVD regarding achievable compression rates, they exhibited poor extrapolation performance when trained on sparse data. This issue could potentially be mitigated by training the autoencoders with additional synthetic data, which would not necessitate costly simulations; however, further research is needed to explore this approach.

## F.2 Additional results

We briefly present additional results from Section 4.4 to demonstrate that the differences in predictions across the various model architectures remain consistent across multiple instances. Figure F.2 depicts the RMSEs of each model from Section 4.4 computed over all 25 trajectories and the spatial domain. The behavior of the models with stability bias can be distinguished from the NODE and PHNN, which suffer from instabilities that result in rapidly increasing errors. While both sPHNN variants and bPHNNs exhibit decreasing errors as the system returns to thermal equilibrium, sPHNNs achieve a peak RMSE nearly an order of magnitude lower, due to their strong inductive bias from enforcing global stability. The sPHNN-LM performs almost identically to the sPHNN initially, while showing slightly larger errors in the convergence to equilibrium. This is due to slight errors in the learned equilibrium position, which can be addressed by increasing the amount of training data, as demonstrated in Section 4.3.

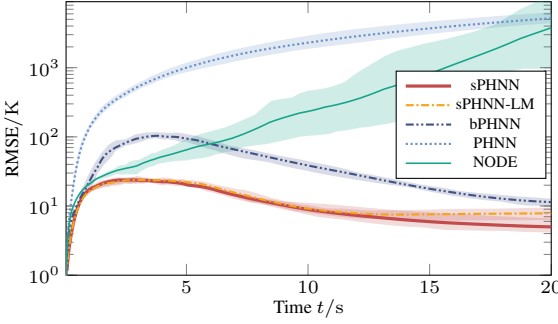

*Figure F.2.* Interquartile mean and range of the RMSEs evaluated with all 25 trajectories of the thermal field data.

