# OpenReview forum: "Stable Port-Hamiltonian Neural Networks"
_NeurIPS.cc/2025/Conference — NeurIPS 2025 poster_

### Official Review · Reviewer_BVE6 · 2025-06-12

**Clarity:** 4
**Significance:** 3
**Originality:** 3
**Rating:** 5
**Confidence:** 3

**Summary:**

This paper tackles the problem of learning stable dynamical systems from data using neural networks. Building on the framework of Lyapunov stability and port-Hamiltonian systems, the authors propose a novel architecture, stable port-Hamiltonian neural networks (sPHNN), in which the system dynamics are modeled using port-Hamiltonian structure. Neural networks are used to learn an energy function and structured matrices representing energy-conserving flows, dissipative effects, and control inputs. The architecture enforces Lyapunov conditions on the learned energy function, thereby guaranteeing asymptotic stability. The proposed model is evaluated on a set of experiments, including both synthetic and real-world datasets, and demonstrates improved stability and data efficiency compared to several baselines.

**Questions:**

1. In Theorem 3.1, isn't coercity for a convex function admitting an uniquer minimizer a standard result in convex analysis? See Rockafeller's Convex Analysis book, Corollary 8.7.1.

2. In line 93, the derivative $\dot{V}(x)$ is scalar-valued. Should it not be referred to as “nonpositive” rather than “negative semi-definite”? Also, I found the second member of the equality chain (2) confusing as $\dot{x}$  is a velocity, implying a trajectory, while the definition $\dot{V}(x)$ is a function of the state and can be defined as a function of $f$, $V$ and $x$ regardless of any trajectory or dynamical system.

3. How is the prediction task evaluated? Are the models used autoregressively, with known or fixed control inputs $u$? More details on this setup would help interpret the results.

4.   If the system is known to admit a Lyapunov function vanishing at $x_0 = 0$, can we expect the learned Hamiltonian to converge to this function under the training dynamics?

5.   What is the interpretation or expected benefit of applying a Hamiltonian structure to systems that are not inherently physical or do not follow Hamiltonian dynamics (e.g., temperature fields)? Could the same stability guarantees not be obtained more simply by enforcing a global Lyapunov function, as done in sNODEs?

6.   The proposed architecture involves gradient computations of the learned Hamiltonian, which may incur additional cost. How does the computational complexity compare to standard NODEs or other baseline methods?

**Ethical Concerns:**

["NO or VERY MINOR ethics concerns only"]

**Final Justification:**

The proposed method is solid and the paper is clearly written. The authors properly addressed my questions. I maintain my positive score of 5.

**Limitations:**

The submission does not discuss some important potential limitations such as the computational cost. See my question 6.

**Quality:**

4

**Strengths And Weaknesses:**

Overall, the paper is very pleasant to read. In spite of minor clairty issues, the method is soundly and clearly presented, and the results are strong.

### Strengths

- The paper is clearly written and mathematically well-grounded. The theoretical framework is well-motivated, and all necessary background is introduced with appropriate references.
- The proposed approach addresses an important and practical problem: learning dynamical models that are both data-efficient and stable, which is particularly relevant in low-data regimes.
- The architecture is original and benefits from strong theoretical foundations. Structuring the model around a learned energy function with interpretable terms (conservative and dissipative) adds physical meaning and interpretability.
- The paper provides a clear and detailed exposition of the model, with Theorem 3.1 nicely summarizing the desirable properties of the architecture.
- The baseline methods are well-documented and chosen appropriately, enabling a fair comparison.
- Experimental results on real-world and high-dimensional synthetic systems show a clear advantage of the proposed method over alternatives.

### Weaknesses

  1. It would be insightful to provide early in the paper an example or figure of a physical system illustrating the need for stability, and what an unconstrained architecture might produce. It would also be nice to introduce an example of port-Hamiltonian system to illustrate the framework of section 2. Typically, a torque-actuated pendulum with friction, where all the terms have a clear physical interpretation, would help ground the abstract formulation of (3). Perhaps a slightly more complex system might also be interesting to expose.

2.   Although the model enforces structure via an interpretable decomposition into conservative and dissipative terms, this structure is defined with respect to a learned energy function. This function is optimized jointly with other parameters to minimize prediction error, and does not necessarily align with any known physical energy.
In the case where the learned system is known to be close to be Hamiltonian, and some prior knowledge on some energy function of the system is available, I am understanding that it could be incorporated in function $f$, and (3) then mimics the system dynamics. When it is not the case though, I am wondering what the interpretation of modeling the dynamics with (3) is. Why should we expect this structure to learn more efficiently, compared to alternative Lyapounov-based learning architectures? See my questions 4. and 5..

3. Evaluation methodology needs clarification.  The experimental setup, particularly how predictions are computed and evaluated, lacks clarity. See question 3.

4.  The paper does not discuss potential downsides of the proposed approach, especially regarding computational cost. The gradient computations involved in the Hamiltonian structure may significantly increase training time.

---

> ### Author Rebuttal · Authors · 2025-07-30
>
> We sincerely thank the reviewer for their thoughtful and constructive feedback, as well as the considerable effort invested in this review. The comments were particularly insightful and raised important points that have helped us improve both the clarity and depth of the manuscript. Below, we respond to each suggestion in detail and outline the corresponding revisions we plan to implement.
>
>
> > It would be insightful to provide early in the paper an example or figure of a physical system [...]
>
> We thank the reviewer for the suggestion and fully agree that such an early example will enhance the accessibility and motivation. We propose to move the spinning rigid body example, especially Equation C.3 and Figure C.1, before Section 3. This example provides a physically interpretable system within the port-Hamiltonian framework and clearly illustrates the benefits of incorporating physically motivated biases when identifying its dynamics.
>
> While learning the dynamics of simple systems using unconstrained methods often does not develop the same level of instability as in more practically relevant scenarios (see, e.g., Sections 4.3 and 4.4), we believe that this example would still be insightful and engaging. In particular, it would help demonstrate the interpretability advantages afforded by the port-Hamiltonian structure.
>
>
> > In Theorem 3.1, isn't coercity for a convex function admitting an uniquer minimizer a standard result in convex analysis? See Rockafeller's Convex Analysis book, Corollary 8.7.1.
>
> While the result might be standard, we’ve still included it in our proof in the appendix to aid in the intuitive, geometric understanding. We will add a suitable reference in the revision.
>
>
> > In line 93, the derivative $\dot V(x)$ is scalar-valued. Should it not be referred to as “nonpositive” rather than “negative semi-definite”? Also, I found the second member of the equality chain (2) confusing as $\dot{x}$ is a velocity, implying a trajectory, while the definition $\dot{V}(x)$ is a function of the state and can be defined as a function of $f$, $V$ and $x$, regardless of any trajectory or dynamical system.
>
> Negative semi-definiteness refers here to the definition of definiteness of a function, which is related to, but distinct from, the definiteness of matrices. Furthermore, the notation $\dot{V}(x)$ denotes the change of the function $V$ along the trajectory $x(t)$, which reduces Equation (2) simply to the application of the chain rule. This notation is standard in the literature about Lyapunov theory (see, for example, H. K. Khalil “Nonlinear systems”, or F. Verhulst “Nonlinear differential equations and dynamical systems”). However, to the best of our knowledge, it is rarely used outside of stability theory.
>
>
> > How is the prediction task evaluated? Are the models used autoregressively, with known or fixed control inputs $u$? More details on this setup would help interpret the results.
>
> All models take the form of an ODE $\dot{x}=f(x,u(t))$, and predictions are generated by numerically integrating this system forward in time using a known input function $u(t)$. This is not an autoregressive setup in the conventional sense, as the models predict state derivatives rather than directly outputting the next state. For our evaluation, the input trajectory is provided beforehand for the duration of the prediction. Nevertheless, the model can be used in a setting where $u(t)$ is supplied in real time, without requiring access to the whole input trajectory in advance.
>
> To clarify this in the manuscript, we will revise line 246 to:
> “Model predictions are generated by rollout with a numerical integrator using the Runge-Kutta scheme Tsit5, with adaptive step size and a known input function $u(t)$.”
>
>
> > If the system is known to admit a Lyapunov function vanishing at x_0=0, can we expect the learned Hamiltonian to converge to this function under the training dynamics?
>
> Lyapunov functions are generally not unique, so one should not expect the learned Hamiltonian to converge to any particular Lyapunov function, even if one is known for the true system. Importantly, for establishing stability properties, only the existence of a Lyapunov function matters - not its specific form. Our model enforces this by constraining the Hamiltonian always to satisfy the Lyapunov conditions, thereby ensuring global stability regardless of whether it matches a known energy function.
> However, since the Hamiltonian not only acts as a Lyapunov function but also generates the port-Hamiltonian dynamics, the best predictions are achieved if the learned Hamiltonian resembles the true system's energy.
>
>
> > What is the interpretation or expected benefit of applying a Hamiltonian structure to systems that are not inherently physical or do not follow Hamiltonian dynamics (e.g., temperature fields)?
>
> Indeed, systems like temperature dynamics are not inherently Hamiltonian in the classical sense, and non-physical systems may lack a physical energy. In such cases, our use of a port-Hamiltonian structure is not necessarily intended to recover a physical Hamiltonian, but rather to enforce a principled, structured decomposition of the dynamics into conservative and dissipative components.
> Compared to models without structure, this decomposition offers interpretability and modularity. Even without physical correspondence, the decomposition can yield meaningful insight into the system’s qualitative behaviour (see, e.g., Figure 1b). Conversely, knowledge about the quantitative behaviour of the system can be integrated into the individual components.
>
>
> > Could the same stability guarantees not be obtained more simply by enforcing a global Lyapunov function, as done in sNODEs?
>
> While sNODEs also provide a stability guarantee, we argue this is not achieved more simply, neither in computational terms nor in practical application. Our reasoning is twofold:
> - **Computational complexity**: sNODEs require numerically evaluating gradients of a learned Lyapunov function and enforcing stability via a projection step. This offers no computational advantage over sPHNNs. In fact, for our numerical example 4.1, sPHNNs were faster to train and to evaluate than sNODEs (see tables below). While we did not focus on runtime optimisation (as the models trained in <2 minutes), these results suggest that sPHNNs are at least as efficient, if not simpler, in practice.
> - **Numerics** The projection step in sNODEs introduces discontinuities in the dynamics (see the discussion at the beginning of Section 4 and in the introduction). These discontinuities can make the dynamics more challenging to integrate numerically, often causing adaptive-step integrators to take smaller steps and, in turn, requiring more model evaluations. Furthermore, in combination with trajectory fitting, we observed failure at training convergence for sNODEs, which excluded them from most of our numerical examples. In contrast, methods like our sPHNN, which don't rely on a projection to guarantee stability, do not suffer from these issues. In cases where the target dynamics are not inherently physical or do not follow Hamiltonian dynamics, one can therefore view the port-Hamiltonian formulation of sPHNNs simply as a way of achieving projection-free stability guarantees.
>
> In summary, even for non-physical systems, sPHNNs offer a structured, projection-free framework for learning stable dynamics, combining theoretical guarantees with practical advantages.
>
>
> > The proposed architecture involves gradient computations of the learned Hamiltonian, which may incur additional cost. How does the computational complexity compare to standard NODEs or other baseline methods?
>
> We agree that a more detailed comparison of the computational performance will enhance the paper.
>
> For training the models, we measured the following average training times:
>
> *Table 1: Representative training times*
> | Training Time (min)              |  sPHNN |  bPHNN |   PHNN |   NODE | sNODE |
> |----------------------------------|-------:|-------:|-------:|-------:|------:|
> | Derivative fitting (Section 4.1) |  0.388 |  1.291 |  0.346 |  0.223 | 1.456 |
> | Trajectory fitting (Section 4.3) | 37.663 | 38.329 | 19.111 | 17.951 |     - |
>
> For inference, we recorded the following times using data from Section 4.3:
>
> *Table 2: Representative evaluation times*
> | Evaluation time (ms)   |   sPHNN |   bPHNN |   PHNN |   NODE |
> |:-----------------------|--------:|--------:|-------:|-------:|
> | Derivative evaluation  |   0.006 |   0.006 |  0.006 |  0.005 |
> | Model integration      |   0.355 |   0.410 |  0.228 |  0.219 |
>
> Here, “Derivative evaluation” refers to the average time taken by each model to compute the state derivative $\dot{x}$ for a given state $x$ and input $u$, which needs to be evaluated for time integration. In contrast, “Model integration” denotes the time required to generate a single trajectory prediction via integration. While the latter better reflects practical use cases, the reported integration time also depends on the integration scheme and chosen step size. Since we use an adaptive step size controller, the step size in turn depends on the learned dynamics, which explains the evaluation differences between sPHNN and PHNN: the latter predicted wrong but smooth dynamics, which may be faster to integrate.
>
> In summary, while the proposed sPHNN introduces some additional computational cost relative to unconstrained NODEs, it performs similarly to or better than baseline models that also provide stability guarantees. Importantly, all models exhibit short training times compared to conventional deep learning approaches, due to their ability to capture complex dynamics with relatively small neural networks. Consequently, computational efficiency typically does not present a bottleneck in practical scenarios.
>
> We will include these results and the accompanying discussion in the appendix of the revised manuscript.

---

> > ### Comment · Reviewer_BVE6 · 2025-08-03
> >
> > Thank you for these clarifications.
> >
> > I think that it is important that the authors clarify the comparison between the proposed method and related approaches in the paper, both from a conceptual (as pointed out by other reviewers) and a computational point of view. The proposed method is interesting, the paper is very well written and the results are solid. I maintain my score.

---

> > > ### Author Response · Authors · 2025-08-06
> > >
> > > Thank you for your comment. We appreciate the confirmation and will ensure the discussed clarifications are incorporated into the final manuscript.

---

### Official Review · Reviewer_n6dB · 2025-06-24

**Clarity:** 3
**Significance:** 3
**Originality:** 2
**Rating:** 5
**Confidence:** 3

**Summary:**

This paper discusses a type of physics guided neural network based on port-Hamiltonian systems, and in particular how global Lyapunov stability can be enforced in such a network. Lyapunov stability is desirable for some problems, because it is often a realistic inductive bias for physical systems, can improve extrapolation, and makes model training more robust.  The model works by representing each of the constituent terms of a port-Hamiltonian system (Hamiltonian, structure matrix, dissipation matrix, and input matrix) with a neural network. The key contribution seems to be the choice of architecture for the Hamiltonian network; by ensuring it is convex (and has a minimum of value 0 at the origin) global stability is enforced. The proposed model is validated on a number of different examples, both with real and synthetic data, and is shown to perform better than alternatives.

**Questions:**

- Please can you more clearly describe the contribution of the proposed model in relation to prior work?
- Can you explain in more detail which real word problems this type of model can be applied to? How easy is it to determine if a real system is globally stable within the region of interest? How well does the model deal with e.g. noisy data?

**Ethical Concerns:**

["NO or VERY MINOR ethics concerns only"]

**Final Justification:**

Authors addressed all my concerns in response, and so I am raising my score to "Accept".

**Limitations:**

yes

**Quality:**

4

**Strengths And Weaknesses:**

## Strengths

- The paper is technically sound to the best of my knowledge (Note: I have not checked the proofs in detail, and do not have a deep knowledge of the theory employed). The main claims made by the paper are adequately supported by the experiments. The experiments are standard for papers in this area.
- The clarity of the paper is excellent. The writing is clear and easy to follow, and the mathematical exposition precise. The paper has a very polished feel, and the figures and plots are well designed and communicate the experimental results well. I enjoyed reading this paper!
- While impact is difficult to judge, the proposed method seems to solve the problem in a more comprehensive way than past work (e.g OnsagerNet).
- The paper is original in the sense that it offers a new combination of existing techniques.

## Weaknesses

- For me, the main weakness of this paper is the lack of clear explanation of the relationship of the proposed method to prior work. This makes it difficult for the reader to understand exactly what the contribution is. A number of related methods are discussed in the introduction, but it is unclear which of these proposed methods could be used to tackle the problems in the experiments section. The only existing method that seems to be used as comparison in the experiments is the sNODE, could other methods be applied here?
- If I am correct in my understanding that the key contribution is constraining the form of the Hamiltonian network, then the contribution of the paper is modest in relation to prior work.

---

> ### Author Rebuttal · Authors · 2025-07-30
>
> We appreciate the reviewer’s thorough evaluation and thoughtful comments. We are glad the clarity and rigour of the paper resonated well. Below, we aim to clarify the specific contributions of our approach, explain the choice of baselines in our experiments, and elaborate on how our method complements and extends existing techniques in the field. We also outline the planned changes in response to the review.
>
>
> > [...] The only existing method that seems to be used as comparison in the experiments is the sNODE, could other methods be applied here?
>
> We want to clarify that the bounded PHNN (bPHNN) baseline model included in all our numerical experiments represents a version of OnsagerNet [Yu 2021], extended to support arbitrary time-dependent inputs to make them more versatile and suitable for our experimental setting.
>
>
> > Please can you more clearly describe the contribution of the proposed model in relation to prior work?
>
> We can differentiate prior work that enforce some notion of stability into projection based methods such as stable NODEs ("learning stable deep dynamics models" [Kolter 2019]) or similar methods [Kojima 2022, Takeishi 2021, Okamoto 2024] and constrained methods for dissipative dynamics such as the OnsagerNet proposed by [Yu 2021]. The former learn unconstrained nominal dynamics and then project them onto the subspace of stable systems, as determined by a concurrently learned Lyapunov function. In contrast, our approach leverages the port-Hamiltonian formalism to formulate the learning problem directly within the subspace of stable dynamics, thereby eliminating the need for projection and avoiding associated numerical challenges, such as discontinuities in the dynamics resulting from the projection. These can hinder the training via trajectory fitting, which is required for modeling dynamics with augmented states. This also effectively excluded stable NODEs from most of our numerical examples, see also the discussion at the beginning of Section 4.
>
> In contrast, constrained methods such as our stable port-Hamiltonian neural networks (sPHNNs) and OnsagerNet (or here bPHNNs)  formulate the learning problem directly in the subspace of stable dynamics, sidestepping the mentioned issues.
> While the structure of the evolution equation of the OnsagerNet is quite similar to sPHNNs, there are several key differences between both approaches:
> - **Inputs**: OnsagerNets do not consider arbitrary, time-dependent external input signals. This is possible with sPHNNs and is even required for most of the numerical examples we present.
> - **Energy sources**: OnsagerNets use state-dependent external forces to model autonomous systems with internal energy sources (necessary for systems with, e.g., limit-cycles or strange attractors). This introduces an ambiguity for decomposing the dynamics into the potential $V$ and the forcing term $f$, potentially weakening the induced physical bias and interpretability. In contrast, for sPHNNs, energy sources necessarily enter through the external inputs $u$, clearly distinguishing between internal dissipative dynamics and external influences.
> - **Stability**: OnsagerNets apply a potential that is bounded from below by zero and coercive (i.e., radially unbounded). Under certain conditions on the forcing term, OnsagerNet guarantees bounded solutions and at least one stable equilibrium, though multiple stable and unstable equilibria may exist. In contrast, sPHNNs enforce global asymptotic Lyapunov stability and guarantee a single equilibrium. While this is more restrictive, it introduces a stronger inductive bias, leading to improved performance in applicable cases, particularly in the small-data regime.
>
>
> > Can you explain in more detail which real word problems this type of model can be applied to?
>
> The proposed model is based on the port-Hamiltonian framework, which is widely used to describe physical systems in various fields, including mechanics, electromechanics, quantum mechanics, control theory, acoustics, fluid dynamics, and thermodynamics. This broad applicability carries over to our model, as demonstrated by our experiments for problems in rigid body dynamics, fluid mechanics, conjugate heat transfer, and thermal diffusion.
>
> Specifically, the stable port-Hamiltonian neural networks (sPHNNs) we introduce can be applied to identifying the dynamics of such systems, provided the system is globally stable. While global stability is a strong assumption, our numerical examples show that it holds in many practically relevant scenarios. In such cases, sPHNNs offer a strong inductive bias, which can significantly improve learning efficiency and generalisation compared to more general-purpose models.
>
> In practical applications, we envision sPHNNs being used as efficient yet accurate surrogate models for complex simulations, particularly in engineering contexts where stability and physical consistency are critical. We demonstrate this applicability in Sections 4.2-4.4.
>
>
> > How easy is it to determine if a real system is globally stable within the region of interest?
>
> Without prior knowledge, determining the stability properties of a given system is a challenging task. While in many cases, even basic physical insights can be helpful, such knowledge may not always be readily available. Under these circumstances, we see our model as a first step in an iterative process: after training an sPHNN, its performance can indicate the validity of the global stability assumption. This approach is low-cost, requiring minimal data, and is fast due to the model’s small network sizes. If performance is insufficient, one can then iteratively explore more general models with weaker inductive biases such as bPHNNs, NODEs, etc..
>
>
> > How well does the model deal with e.g. noisy data?
>
> In our numerical experiments, the model demonstrated strong robustness due to the incorporation of the global stability constraint. This was especially evident with respect to variations in the initial trainable parameters and the quantity of training data. While the real-world measurement data employed in Section 4.2 is noisy, it has a relatively large signal-to-noise ratio.
>
> To more directly assess the model’s robustness to noise, we conducted an additional experiment based on the setup presented in Section 4.3. We have trained the models using $n_A=3$ augmented dimensions and $n_D=2$ training trajectories, introducing zero-mean Gaussian noise for both the input $T_{oven}$ and the outputs $T_A$ and $T_B$. The noise amplitudes were varied between 5% and 25% of the original signals’ standard deviations. The table below reports the average root mean squared error (RMSE) over 20 model instances per model type, evaluated on noise-free test data.
>
> *Table 1: Test RMSE*
> | Model   |   5% noise |   15% noise |   25% noise |
> |:--------|-----------:|------------:|------------:|
> | sPHNN   |      1.170 |       2.595 |       3.177 |
> | bPHNN   |      1.572 |       2.828 |       5.231 |
> | PHNN    |     28.233 |      25.631 |      24.224 |
> | NODE    |      3.577 |       3.482 |       4.362 |
>
> To isolate the effect of noise and eliminate generalisation error, we also evaluated the RMSE of the models trained on noisy data on the noise-free training data:
>
> *Table 2: Training RMSE*
> | Model   |   5% noise |   15% noise |   25% noise |
> |:--------|-----------:|------------:|------------:|
> | sPHNN   |      0.332 |       0.854 |       1.354 |
> | bPHNN   |      0.500 |       1.049 |       2.207 |
> | PHNN    |      2.828 |       6.750 |       5.779 |
> | NODE    |      0.993 |       1.974 |       2.423 |
>
> While all models demonstrate a degree of robustness to noise, the sPHNN consistently outperforms the baselines on both the training and test sets. This robustness can be partially attributed to the training process: the integration step involved in trajectory fitting inherently acts as a low-pass filter, effectively smoothing the learned dynamics. As a result, high-frequency Gaussian noise is attenuated, which helps prevent overfitting and improves generalisation.
>
> We will incorporate the additional results and corresponding discussion in Section 4.3 of the revised manuscript.

---

### Official Review · Reviewer_1bj2 · 2025-07-02

**Clarity:** 3
**Significance:** 3
**Originality:** 2
**Rating:** 3
**Confidence:** 3

**Summary:**

This paper aims to incorporate physical prior into dynamical system learning. The authors introduce a framework named stable port-Hamiltonian neural networks. This framework aims to previous the physical biases while ensuring global Lyapunov stability. They also establish the requirement for stability of port-Hamiltonian systems and utilize this for model design. The authors evaluate the proposed method on different systems in comparison to NODEs, PHNNs, and bounded PHNNs (bPHNNs).

**Questions:**

Please see weaknesses.

**Ethical Concerns:**

["NO or VERY MINOR ethics concerns only"]

**Final Justification:**

More baselines are needed.

**Limitations:**

Yes

**Quality:**

2

**Strengths And Weaknesses:**

Strengths

1. This paper is well organized and clearly written.

2. The technical details are also easy to follow.

3. The authors evaluate the proposed method on different systems to validate the effectiveness of the proposed method.

Weaknesses

1. The major concern is that the compared methods are out-of-date and limited. I do suggest that authors should include more state-of-the-art and up-to-date methods in performance comparison given that there are so many works for dynamical system modeling.

2. It seems that all the data is generated by simulation. Are there any real-world datasets to validate the effectiveness of the proposed method?

3. The difference between Stable Port-Hamiltonian Neural Networks and Stable NODEs should be carefully discussed to show the novelty of the proposed method.

4. "Why port-Hamiltonian dynamics?" seems not to be the contribution of the paper. This part should move to the preliminary section instead.

---

> ### Author Rebuttal · Authors · 2025-07-30
>
> We appreciate the reviewer’s thoughtful comments and the opportunity to clarify the scope and contributions of our work. Below, we address each point in detail, explaining our methodological choices and planned revisions to strengthen the manuscript.
>
>
> > The major concern is that the compared methods are out-of-date and limited. I do suggest that authors should include more state-of-the-art and up-to-date methods in performance comparison given that there are so many works for dynamical system modeling.
>
> To include further baselines, we would appreciate it if the reviewer could provide more details on which methods are considered as state-of-the-art and should be included.
> Certainly, deep learning architectures such as transformers have shown great potential for time series forecasting in recent years, but are computationally demanding, data hungry, and requiring a lot of fine-tuning of the architectures (see, e.g., Chen et al., “A Closer Look at Transformers for Time Series Forecasting: Understanding Why They Work and Where They Struggle”). Thus, we focus on comparing to methods that model dynamic systems as ODEs, which is close to classical, physics-based modeling and allows to enforce properties such as thermodynamic consistency and stability, which we deem as highly beneficial for obtaining reliable predictions. In this context, we regard (stable) NODEs and energy-based concepts such as PHNNs (or similarly GENERIC, which exhibits similar performance, see Urdeitx et al., “A comparison of single and double generator formalisms for thermodynamics-informed neural networks”.) as reasonable baselines.
>
>
> > It seems that all the data is generated by simulation. Are there any real-world datasets to validate the effectiveness of the proposed method?
>
> The data used in the cascaded tanks example in Section 4.2 originates from measurements of a real-world fluid level control system. This setup reflects an actual physical process and involves real sensor data.
>
> Although the data in Section 4.3 is generated through simulation, it is based on a high-fidelity multiphysics model that closely captures the physical processes of an actual thermal food processing system, where temperatures are measurable using standard temperature sensors.
>
> To further address the potential impact of measurement noise, which may be absent from simulated data, we have included additional evaluations showing that the proposed method remains highly robust to random additive perturbations in the training data. For further details, please see our response to Reviewer n6dB.
>
>
> > The difference between Stable Port-Hamiltonian Neural Networks and Stable NODEs should be carefully discussed to show the novelty of the proposed method.
>
> The main difference between our approach and stable NODEs ("learning stable deep dynamics models" [Kolter 2019]) or similar methods [Kojima 2022, Takeishi 2021, Okamoto 2024] is that our approach is projection-free. The named methods learn unconstrained nominal dynamics and then project them onto the subspace of stable systems, as determined by a concurrently learned Lyapunov function. In contrast, our approach leverages the port-Hamiltonian formalism to formulate the learning problem directly within the subspace of stable dynamics, eliminating the need for projection and avoiding associated numerical challenges. Additionally, the port-Hamiltonian formulation of our model offers interpretability (e.g., the split into conservative and dissipative dynamics) that other approaches lack.
>
> As we show with the spinning rigid body experiment in Section 4.1 (see Fig. 2 and C.1), the sNODEs show stable behavior (they converge to 0 energy and angular velocities), but their accuracy and variance are much worse than the proposed sPHNNs.
>
> To complement the discussion of the above points in the introduction, we will revise line 342 to read: “Additionally, the approach ensures global asymptotic stability of the identified dynamics without requiring projection, by constraining the Hamiltonian to be a convex, positive definite Lyapunov function.”
>
>
> > "Why port-Hamiltonian dynamics?" seems not to be the contribution of the paper. This part should move to the preliminary section instead.
>
> We agree with the reviewer and will integrate Section 3.1 (“Why port-Hamiltonian dynamics”) into the background Section 2.2.

---

> > ### Comment · Reviewer_1bj2 · 2025-08-06
> >
> > My concerns are partially addressed. There are many neural ODE models that outperform the vanilla ODE models, which may serve as baseline models.

---

> > > ### Author Response · Authors · 2025-08-06
> > > **Discussion of baselines**
> > >
> > > We thank the reviewer for raising the point regarding baseline models. While the comment suggests that there are “many neural ODE models” that may outperform standard NODEs, no specific baselines were mentioned. We have surveyed the literature and considered several enhanced neural ODE variants as possible comparators. These include Augmented NODEs (Dupont et al., 2019), Latent ODEs (Rubanova et al., 2019), and Neural CDEs (Kidger et al., 2020), among others.
> > >
> > > However, our work specifically focuses on learning **stable** dynamics as a means to improve generalisation and reduce the required training data. As such, our most relevant baselines are other methods that explicitly enforce stability or structure, such as sNODEs or OnsagerNet. Many popular NODE variants do not provide stability guarantees or model the same physical inductive biases we target. We believe this justifies our choice of our primary baselines.
> > >
> > > Additionally, we would like to clarify that the neural ODE baseline presented in our manuscript already incorporates enhanced modelling strategies: it uses augmented states in Sections 4.2 and 4.3, and a latent space formulation in Section 4.4. As such, it goes beyond a vanilla neural ODE and - to the best of our knowledge - aligns with current best practices for expressive ODE modelling.
> > >
> > > That said, we are open to including additional baselines in a future version, especially if the reviewer has specific models in mind that align with our focus on stability and physical structure.

---

### Official Review · Reviewer_FvJb · 2025-07-03

**Clarity:** 4
**Significance:** 2
**Originality:** 2
**Rating:** 5
**Confidence:** 5

**Summary:**

This work concerns an introduction of new neural network architecture for data-driven dynamical system identification based on port-Hamiltonian systems. The architecture is based on the port-Hamiltonian system formulation, in which (1) the structure matrix and the dissipation matrix are each modeled by feed forward neural networks and their structure is enforced (skew-symmetry and symmetric positive semi-definite), (2) the port-Hamiltonian is modeled by a fully input convex neural network (FICNN). The authors show that this new formulation is asymptotically stable, based on a theorem developed in the work. The new model is tested on a few dissipative systems, and shows improved asymptotic behavior as well as superior overall accuracy.

**Questions:**

- Can this framework be extended to parametrized systems?

- Why is the Lyapunov function introduced, if it is not required in the Theorem? It appears the function is only necessary in the proof?

- Is there another baseline comparison beyond deep learning models (that also incorporate classical methods like SINDy: Lee, Kookjin, Nathaniel Trask, and Panos Stinis. "Structure-preserving sparse identification of nonlinear dynamics for data-driven modeling." Mathematical and Scientific Machine Learning. PMLR, 2022.)?

**Ethical Concerns:**

["NO or VERY MINOR ethics concerns only"]

**Final Justification:**

The reviewers have provided corrections and have addressed my confusion regarding the knowledge of the equilibrium point $x_0$ and provided evidence that the methodology can still apply when $x_0$ is unknown. The authors agreed to incorporate clarifying comments regarding the key scope of the paper.

**Limitations:**

- The explicit specification using $\mathbf{x}_0$ should be elaborated upon. More examples where equilibrium points are not known a priori would be enlightening.

**Paper Formatting Concerns:**

Did not notice any.

**Quality:**

3

**Strengths And Weaknesses:**

Strengths

- The presentation is clear and straightforward. The new architecture as well as the main theorem is explained clearly.

- The specific formulation of the port-Hamiltonian is new to this reviewer (although the use of input convex neural networks for Hamiltonian systems is not new).

Weaknesses

- The developed theorem requires a specification of a special point in the state space $\mathbf{x}_0$. This requirement is a bit peculiar, given that generally stability conditions are often formulated without specifying special points. This point is not sufficiently elaborated upon in the paper; how do you select this point in general situations? Does one need to retrain the model if the point needs to be changed?

---

> ### Author Rebuttal · Authors · 2025-07-30
>
> We thank the reviewer for the insightful questions and the opportunity to clarify important aspects of our approach. Below, we address each point, providing explanations and outlining planned revisions to improve clarity and completeness in the manuscript.
>
>
> > The developed theorem requires a specification of a special point in the state space $x_0$. [...] This point is not sufficiently elaborated upon in the paper; how do you select this point in general situations?
>
> The proposed model incorporates the equilibrium position $x_0$​ directly into its architecture, but its value does not need to be known in advance.
> In the theoretical parts of the paper, such as Theorem 3.1, we assume $x_0=0$ to simplify notation. This is without loss of generality and standard practice in stability theory literature [Verhulst 1990, Khalil 2015]. However, in a general application where $x_0$ is unknown, we treat it as a trainable parameter that is optimised during training. This way, arbitrary equilibrium positions can be inferred from observations of the system. We demonstrate this in section 4.2 for the cascaded tanks example, where $x_0$ is learned from the measurement data.
> While learning $x_0$ is possible, we argue that for many relevant and globally stable systems the equilibrium can be inferred a priori from physical reasoning, as demonstrated in our other numerical examples. Including this knowledge as an inductive bias into the model can significantly improve model quality.
>
> To emphasise the above discussion more explicitly, we will revise line 197 to read: “Fixing $x_0$ to the true equilibrium position introduces an additional inductive bias. While this information can often be obtained from prior physical knowledge, it may not always be available. In such cases, the equilibrium $x_0$ can be inferred directly from data by treating it as a trainable parameter during model training.
>
>
> > Does one need to retrain the model if the point needs to be changed?
>
> For a specific, non-parametric dynamic system, the equilibrium point is fixed and does not change. As described above, it can be considered as a trainable parameter and thus be identified from data.
> However, for a parameterised system, also $x_0$ could depend on the parameter set $\mu$. In ongoing work, we are extending the presented approach to parameterised systems, allowing parameter-dependent equilibrium points $x_0(\mu)$, see also the next answer below.
>
>
> > Can this framework be extended to parametrized systems?
>
> Yes. By including a set of parameters $p$ as inputs to the neural networks describing the Hamiltonian $\mathcal{H}$ and the matrices $\boldsymbol{J}$, $\boldsymbol{R}$, and $\boldsymbol{G}$, the parametric dependence of the dynamics can be learned from data. The equilibrium position $x_0$ can similarly be modelled as a neural network function of $p$.
> To ensure sufficient flexibility and to avoid restricting $\mathcal{H}(x, p)$ to be convex in $p$, partially input convex neural networks (PICNNs) [Amos et al., 2017] can be used to parameterise the Hamiltonian. We are currently developing this extension and could include it in the final version for one example, such as the spinning rigid body, which can be parameterised by its principal moments of inertia $I_1, I_2, I_3$ or a damping coefficient $\mu$.
>
>
> > Why is the Lyapunov function introduced, if it is not required in the Theorem? It appears the function is only necessary in the proof?
>
> We acknowledge that the dependency of the theorem on Lyapunov theory and especially Lyapunov functions is rather implicit and subtle. To address this, we will change line 155 from “Then, the system in Equation (5) has a stable equilibrium at $x(t) = 0$, and all solutions are bounded.” to “Then, the Hamiltonian $\mathcal{H}$ is a suitable Lyapunov function for showing stability of the equilibrium at $x(t)=\boldsymbol{0}$, and all solutions are bounded.”
> With this change we aim to clarify the connection to Lyapunov theory and improve the intuition behind the theorem.
>
>
> > Is there another baseline comparison beyond deep learning models (that also incorporate classical methods like [structure-preserving] SINDy: [...])?
>
> While we have not made a direct comparison with structure-preserving SINDy yet, we expect that the method will struggle with complex dynamics in higher-dimensional settings (in the mentioned reference, the examples have only up to 3 states). In particular, ensuring a convex Hamiltonian or energy for achieving stability will be difficult with SINDy, as multivariate polynomials are generally non-convex (e.g., $x^2 y^2$ is not convex). In our opinion, input-convex NNs are much more versatile for this purpose.

---

> > ### Comment · Reviewer_FvJb · 2025-08-05
> >
> > Thank you for addressing my questions. I have a few follow-up comments.
> >
> > > While learning $x_0$ is possible, we argue that for many relevant and globally stable systems the equilibrium can be inferred a priori from physical reasoning, as demonstrated in our other numerical examples. Including this knowledge as an inductive bias into the model can significantly improve model quality.
> >
> > > For a specific, non-parametric dynamic system, the equilibrium point is fixed and does not change.
> >
> > I'm a bit confused by these statements: I would think that in general it is not easy to ascertain whether the equilibria are, for cases where the port-Hamiltonian is complicated. Is not the one goal of studying these systems to discover the behavior near the equilibria even if they are not known a priori? I will appreciate some clarification on this.
> >
> > It seems to me then, the scope of this paper is a system where (1) there is a single equilibrium and (2) that equilibrium is known. Can the authors make this more clear in the introduction? I think it will aid the readers grasp the main theme easier.

---

> > > ### Author Response · Authors · 2025-08-06
> > > **Clarification regarding unknown equilibrium positions**
> > >
> > > Thank you for the follow-up and for engaging in this active discussion. We appreciate the feedback and the opportunity to provide further clarification.
> > >
> > >
> > > ## Clarifying the scope: Known vs. learned equilibria
> > > We agree that understanding system behaviour near equilibria is an important goal in dynamical systems analysis. However, we would like to emphasise that our approach **does not require** prior knowledge of the equilibrium for this. The normalisation procedure we describe in Equation 8, essentially only reparameterizes the convex Hamiltonian $\mathcal{H}$ such that the equilibrium state $x_0$ becomes an explicit parameter of our network, instead of being implicitly described by all the weights and biases in the input convex neural network. This enables the proposed architecture to accommodate both scenarios:
> > >
> > > - When the equilibrium position $x_0$ is **known a priori**, it can be incorporated into the model to introduce a meaningful inductive bias.
> > > - When $x_0$ is **unknown**, it is treated as a trainable parameter and learned directly from data.
> > >
> > > To avoid confusion, we will clarify in the introduction that the proposed method supports both cases and highlight this flexibility as one of its strengths.
> > >
> > >
> > > ## On physical intuition for equilibrium states
> > > While we acknowledge that some systems may have complex port-Hamiltonian structures, we argue that in many practical, globally stable systems, the equilibrium state can indeed be inferred based on physical intuition or conservation laws.
> > > For example:
> > >
> > > - In the cascaded tanks system (Section 4.2), the complex dynamics of outflow are nonlinear and depend on geometry and fluid properties. Yet, it is clear that in the absence of input flow, the equilibrium state is an empty tank, irrespective of the dynamic complexity.
> > > - A similar argument can be applied to thermal systems (such as in Sections 4.3 and 4.4), where the thermal behaviour can be arbitrarily complex. Still, the equilibrium state corresponds to a homogeneous temperature distribution, which can often be inferred from the boundary conditions.
> > >
> > > We will integrate these arguments more explicitly into the manuscript to clarify that complex dynamics do not preclude a clear a priori identification of equilibria in many real-world systems.
> > >
> > >
> > > ## Learning the equilibrium state
> > > To demonstrate that our model can infer the system dynamics even if the equilibrium is not available a priori, we have applied the sPHNN-LM (“learnable minimum”, introduced in Section 4.2) to the data from Sections 4.1 and 4.3 as well. In this variant, the equilibrium parameter $x_0$ is initialised randomly and optimised during training, meaning no prior knowledge about the equilibrium is provided to the model. After training, the inferred equilibrium can be directly read from the learned parameter.
> > >
> > > **Data from Section 4.1**: For the spinning rigid body data (Section 4.1), the mean distance between the inferred $x_0$ and the actual equilibrium position of the system is 0.010228 (interquartile mean). We thus have recovered the actual equilibrium position with high accuracy. The predictive performance of sPHNN-LM is nearly identical to that of the sPHNN with fixed $x_0$.
> > >
> > > **Data from Section 4.3**: The table below shows the distances of the learned $x_0$ from the true equilibrium and the test RMSE values for the sPHNN-LM model trained with the data from Section 4.3. While the inferred $x_0$ is less accurate in this setting, the predictive errors remain close to those of the sPHNN with known equilibrium. This result is expected, as most training trajectories lie far from equilibrium, making the performance less sensitive to the precise value of $x_0$.  Still, the mean of the inferred equilibrium values across all training runs is 277.25 K, which is remarkably close to the true equilibrium of 279.15 K. These results show that our model can reliably estimate the equilibrium from data alone. Nevertheless, comparing sPHNN-LM to sPHNN test errors confirms that incorporating prior knowledge of $x_0$, when available, can help to improve performance.
> > >
> > > *Table 1: Interquartile mean of $x_0$ error and test RMSE in Kelvin for sPHNN-LM for data from Section 4.3.*
> > > | Augmentation $n_A$ |     0 |      1 |      2 |      3 |
> > > |:-------------------|------:|-------:|-------:|-------:|
> > > | Minimum distance   | 8.765 | 15.894 | 23.318 | 17.134 |
> > > | Test RMSE          | 2.778 |  2.919 |  3.230 |  2.306 |
> > >
> > > We will incorporate this extended discussion and the above results in the final manuscript.

---

> > > > ### Comment · Reviewer_FvJb · 2025-08-06
> > > >
> > > > > We will integrate these arguments more explicitly into the manuscript to clarify that complex dynamics do not preclude a clear a priori identification of equilibria in many real-world systems.
> > > >
> > > > I'm afraid this does not clarify the point I was trying to make. Yes, there are likely equilibria that can be trivially identified, but that does not mean that _all_ equilibria in such systems are known a priori. This brings us back to the situation where these equilibria would need to identified and learned.
> > > >
> > > > I appreciate the efforts to present new results regarding learning the equilibria, but I think the results show that learning $x_0$ warrants a separate investigation -- it does not appear as clear cut to me.
> > > >
> > > > I want to make clear I am not being critical of the overall work: I am saying that the results of the paper support the conclusion that sPHNN works well in the case (1) there is a single equilibrium and (2) that equilibrium is known. Whether the method works in more general settings where there can be multiple or unknown equilibrium appears to remain very much open was not thoroughly studied in this paper in my view. Again, I am not saying the authors should do this work here: Since the authors did not conduct serious experiments focused on this, it is my view that it would be appropriate to state the scope of this work clearly. I think doing so would greatly improve my opinion of this work!

---

> > > > > ### Author Response · Authors · 2025-08-07
> > > > >
> > > > > We would like to thank the reviewer for the clarification and continued engagement.
> > > > >
> > > > > We largely agree with the assessment. While the scope of our method does encompass learning dynamics in both settings (with known and unknown equilibrium positions), the numerical experiments in the paper so far are primarily focused on the case where the equilibrium is known.
> > > > >
> > > > > That said, we do include examples with the sPHNN-LM variant in Section 4.2, demonstrating that equilibrium positions can be learned from data. As mentioned in the previous comment, we would like to include the sPHNN-LM variant also in the experiments 4.1 and 4.3 in the final version. These results show that with increasing training data (c.f. Fig. 4c), sPHNN-LM performs on par with the fixed-equilibrium variant (sPHNN), suggesting that learning the equilibrium is both feasible and effective.
> > > > >
> > > > > We will revise the introduction and conclusion as suggested to reflect the scope of the work better. In particular, we propose changing line 66 to read: “_We demonstrate that the approach can infer equilibrium positions from data, though our numerical evaluations largely focus on one of the key features of the approach: the ability to incorporate explicit knowledge of the equilibrium directly into the model architecture._”
> > > > >
> > > > > We hope this clarification helps communicate our intentions more clearly and appreciate the reviewer’s constructive suggestions.

---

> > > > > > ### Comment · Reviewer_FvJb · 2025-08-07
> > > > > >
> > > > > > Thank you for the replies, I will raise my score.

---

### Note · Authors · 2025-08-15

We thank all reviewers for their constructive feedback and the valuable discussions. As a final remark, we briefly summarise the key revisions planned for the manuscript.

- Clarify in the introduction and conclusion that while our method supports both known and unknown equilibria, the numerical experiments primarily target cases with a known equilibrium, highlighting the benefits of incorporating this knowledge into the model architecture.
- Move the spinning rigid body section to act as an early, illustrative physical example to motivate the port-Hamiltonian framework and enhance accessibility.
- Integrate Section 3.1 (“Why port-Hamiltonian dynamics”) into the background Section 2.2.
- Expand the discussion of differences to related work, in particular stable NODEs, OnsagerNet, emphasising the projection-free stability guarantee and interpretability benefits.
- Add the new results for the sPHNN-LM variant in Sections 4.1 and 4.3 to demonstrate that equilibria can be inferred from data.
- Include the additional robustness-to-noise experiments and a detailed computational cost comparison.
- Improve clarity on the role of the Lyapunov function in our theorem and on the prediction setup.

We believe these changes will strengthen the clarity, scope, and impact of the paper.

---

### Decision · Program_Chairs · 2025-09-17

**Decision:**

Accept (poster)

**Comment:**

Stable port-Hamiltonian neural networks enforce global Lyapunov stability via a convex Hamiltonian parameterization, combining physical interpretability with inductive bias. Strengths are a clear theoretical foundation, projection-free stability guarantees, and solid empirical results. After an extensive discussion period, the authors managed to convince three reviewers fully, who are now clearly in favor of acceptance, highlighting clarity, rigor, and practical relevance. One reviewer questioned the choice of baselines and novelty but did not engage further after the authors’ detailed clarifications, which I find convincing. I recommend acceptance as a poster.